# Dataset Distillation via Knowledge Distillation: Towards Efficient Self-Supervised Pre-training of Deep Networks

**Siddharth Joshi, Jiayi Ni, Baharan Mirzasoleiman**
Department of Computer Science,
University of California, Los Angeles
`sjoshi804@cs.ucla.edu, nijiayi1119626@g.ucla.edu, baharan@cs.ucla.edu`

## Abstract

Dataset distillation (DD) generates small synthetic datasets that can efficiently train deep networks with a limited amount of memory and compute. Despite the success of DD methods for supervised learning, DD for self-supervised pre-training of deep models has remained unaddressed. Pre-training on unlabeled data is crucial for efficiently generalizing to downstream tasks with limited labeled data. In this work, we propose the first effective DD method for SSL pre-training. First, we show, theoretically and empirically, that naïve application of supervised DD methods to SSL fails, due to the high variance of the SSL gradient. Then, we address this issue by relying on insights from knowledge distillation (KD) literature. Specifically, we train a small student model to match the representations of a larger teacher model trained with SSL. Then, we generate a small synthetic dataset by matching the training trajectories of the student models. As the KD objective has considerably lower variance than SSL, our approach can generate synthetic datasets that can successfully pre-train high-quality encoders. Through extensive experiments, we show that our distilled sets lead to up to 13% higher accuracy than prior work, on a variety of downstream tasks, in the presence of limited labeled data. Code at `https://github.com/BigML-CS-UCLA/MKDT`

## 1 Introduction

Dataset distillation (DD) aims to generate a very small set of synthetic images that can simulate training on a large image dataset, with extremely limited memory and compute (Wang et al., 2018). This facilitates training models on the edge, speeds up continual learning, and provides strong privacy guarantees (Kim et al., 2022; Cazenavette et al., 2022; Sajedi et al., 2023; Dong et al., 2022). As a result, there has been a surge of interest in developing better DD methods for training neural networks in a supervised manner. However, in many applications, very few labeled example are available. In this case, supervised models often fail to generalize well. Instead, models are pre-trained, using self-supervised learning (SSL), on a large amount of *unlabeled* data and then adapted to the downstream task using the limited labeled data by training a linear classifier using the labeled examples of each downstream task (linear probe). Remarkably, Chen et al. (2020) showed that SSL pre-training, followed by linear probe, can outperform Supervised Learning (SL) by nearly 30% on ImageNet (Deng et al., 2009) when only 1% of labels are available. More impressively, by only training the linear layer (linear probe), SSL pre-training is able to generalize to a variety of downstream tasks nearly as effectively as full fine-tuning on the downstream task, for a fraction of the cost. Thus, SSL pre-training's benefits are invaluable in today's modern ML ecosystem, where unlabeled data is plentiful and it is essential to generalize to a plethora of downstream tasks, effectively and efficiently. With the datasets for SSL being far larger than those for SL, the computational and privacy benefits of dataset distillation for SSL are even more important than they are for SL. Nevertheless, DD for SSL pre-training has remained an open problem.

DD for SSL is, however, very challenging. One needs to ensure that pre-training on the synthetic dataset, distilled from unlabeled data, results in a encoder that yields high-quality representations for a variety of downstream tasks. Existing DD methods for SL generate synthetic data by matching gradients (Zhao et al., 2021; Zhao & Bilen, 2021; Lee et al., 2022) or trajectory (Cazenavette et al., 2022; Du et al., 2023; Cui et al., 2022) of training on distribution of real data (Zhao & Bilen, 2023;

Sajedi et al., 2023; Wang et al., 2022a), or meta-model matching by generating synthetic data such that training on it achieves low loss on real data (Wang et al., 2018; Nguyen et al., 2021; Loo et al., 2022; Zhou et al., 2022). Among them, gradient and distribution matching methods heavily rely on labels and will suffer from representation collapse otherwise. Hence, they are not applicable to SSL DD. Very recently, Lee et al. (2023) applied meta-model matching to generate synthetic examples for SSL pre-training, and evaluated its performance by fine-tuning the entire pre-trained model on the large labeled downstream datasets. However, we show that SSL pre-training on these distilled sets does not provide any advantage over SSL pre-training on random real examples.

In this work, we address distilling small synthetic datasets for SSL pre-training via trajectory matching. First, we show, theoretically and empirically, that naïve application of trajectory matching to SSL fails, due to the high variance of the gradient of the SSL loss. Then, we rely on insights from *knowledge distillation* (KD) to considerably reduce the variance of SSL trajectories. KD trains a smaller student network to match the predictions of a larger teacher network trained with supervised learning (Hinton et al., 2015). In doing so, the student network can match the performance of the larger teacher model.

Here, we apply KD for SSL by training a student encoder to match the *representations* of a larger teacher encoder trained with SSL. Then, we propose generating synthetic data for SSL by Matching Knowledge Distillation Trajectories (MKDT). Crucially, as the KD objective for training the student model has considerably lower variance, it enables generating higher-quality synthetic data by matching the lower-variance trajectories of the *student* model. As a result, the encoder can learn high-quality representations from the synthetic data. We also provide theoretical and empirical evidence showing that KD trajectories are indeed lower variance than SSL trajectories and that this lower variance enables effective dataset distillation for SSL.

Finally, we conduct extensive experiments to validate the effectiveness of our proposal MKDT for SSL pre-training. In particular, we distill both low resolution (CIFAR10, CIFAR100) and larger, high resolution datasets (TinyImageNet) down to 2% and 5% of original dataset size and show that, across various downstream tasks, MKDT distilled sets outperform all baselines by up to 13% in the presence of limited labeled data. Moreover, we confirm that the datasets distilled with smaller ConvNets can transfer to architectures as large as ResNet-18. Finally, we demonstrate that MKDT is effective across SSL algorithms (BarlowTwins (Zbontar et al., 2021) and SimCLR (Chen et al., 2020)).

## 2 RELATED WORK

### 2.1 DATASET DISTILLATION

There has been a large body of recent work on dataset distillation for supervised learning. These techniques can be broadly characterized into meta-model matching, gradient matching, distribution matching and trajectory matching (Sachdeva & McAuley, 2023).

**Meta-model Matching** Meta-model matching generates synthetic data such that a model trained on the synthetic dataset achieves low training loss on the real dataset (Wang et al., 2018). The traditional meta-model matching approach is computation and memory inefficient as it requires solving a bi-level optimization problem. Thus, several methods (Nguyen et al., 2021; Loo et al., 2022; Zhou et al., 2022) have been proposed to solve the inner-optimization problem in closed form with kernel ridge regression.

**Gradient Matching** Gradient matching generates synthetic data by matching the gradient of a network trained on the original dataset with the gradient of the network trained on the synthetic dataset (Zhao et al., 2021; Zhao & Bilen, 2021; Lee et al., 2022). Gradient-matching is done for each class separately, otherwise optimizing the synthetic images to match gradients is not possible (Zhao et al., 2021). As a result, these methods require labels to be applicable.

**Matching Training Trajectories (MTT)** MTT, first proposed by Cazenavette et al. (2022), generates synthetic data by matching the training trajectories of models trained on the real dataset with that of the synthetic dataset. Cui et al. (2022) reduced the memory footprint of MTT and Du et al. (2023) minimized the accumulated error in matching trajectories by distilling flatter trajectories.

**Distribution Matching** Distribution matching generates synthetic data by directly matching the distribution of synthetic dataset and original dataset. One line of work does so by minimizing the

maximum mean discrepancy (MMD) between the representations of the synthetic and real data using a large pool of feature extractors (Zhao & Bilen, 2023; Sajedi et al., 2023; Wang et al., 2022b). For these methods, distilling examples **per class** is essential, as without labels, the models trained on the synthetic data suffer from representation collapse and cannot learn any discriminative features (Zhao & Bilen, 2023). More recently, (Yin et al., 2024; Zhou et al., 2024; Shao et al., 2024) apply ideas from *data-free knowledge distillation*Lopes et al. (2017) to match the distributions of synthetic and real images using the batch norm statistics of models trained on the full data. While these methods do not distill per class, the distillation loss relies on the labels of the data and is essential to distill data preserving class-discriminative features.

**Dataset Distillation for SSL** Very recently, KRR-ST (Lee et al., 2023) applied the kernel-based meta-model matching to distillation for SSL. However, kernel ridge regression, with a relatively unchanging encoder as the kernel function, prevents distilling synthetic data that is useful for training the encoder effectively. We empirically confirm that the encoder learnt by pre-training on these generated examples cannot outperform encoder learnt directly using SSL pre-training on random real images. While KRR-ST also uses a MSE loss to representations instead of directly performing SSL, they claim to do so to mitigate the **bias** of bi-level optimization in meta-model based matching for SSL. In MKDT, we instead, use the knowledge distillation loss of MSE to representations of a larger teacher model to reduce the high **variance** of SSL gradients, and thus lower variance trajectories enable trajectory matching.

MTT is another dataset distillation method that is agnostic to the labels, and hence can be potentially applied to SSL. Nevertheless, application of MTT to SSL has not been explored before. In our work, we show that that naïve application of MTT to SSL yields poor performance. Then, we propose a method that leverages knowledge distillation to enable effective dataset distillation for SSL.

## 2.2 Data-efficient Learning via Subset Selection

Another line of work that enables data-efficient learning is selecting subsets of training data that generalize on par with the full data. This has been extensively studied for supervised learning (Coleman et al., 2020; Toneva et al., 2019; Paul et al., 2021; Mirzasoleiman et al., 2020; Yang et al., 2023). At a high level, these works show that difficult-to-learn examples with a higher loss or gradient norm or uncertainty benefit SL the most. More recently, SAS (Joshi & Mirzasoleiman, 2023) has been proposed for selecting subsets of data for self-supervised contrastive learning (CL). Interestingly, the most beneficial subsets for SL are shown to be least beneficial for self-supervised CL. We use SAS as a baseline and show that the synthetic data distilled by our method can outperform training on these subsets.

## 2.3 Knowledge Distillation

Knowledge distillation (KD) is a technique used to transfer knowledge from a large teacher model to a smaller student model, with the aim of retaining high performance with reduced complexity (Hinton et al., 2015). For supervised learning, some techniques align the student's outputs with those of the teacher (Hinton et al., 2015), while others concentrate on matching intermediate features (Romero et al., 2015), attention maps (Zagoruyko & Komodakis, 2017) or pairwise distances between examples (Park et al., 2019). Recent works have adapted KD for SSL models (Passalis & Tefas, 2018; Chen et al., 2017; Koohpayegani et al., 2020; Yu et al., 2019). DarkRank (Chen et al., 2017) approaches KD for SSL as a rank matching problem between teacher and student encoders. PKT (Passalis & Tefas, 2018) and Compress (Koohpayegani et al., 2020) model the similarities in data samples within the representation space as a probability distribution, aiming to align these distributions between the teacher and student encoders. Yu et al. (2019) introduced the concept of minimizing the Mean Squared Error (MSE) between the representations of student and teacher encoders. In this work, we rely on KD to enable effective dataset distillation for SSL.

## 3 Problem Formulation

Consider a dataset $\mathcal{D}_{\mathsf{real}} = \{\boldsymbol{x}_i\}_{i \in [n]}$ of $n$ unlabeled training examples drawn i.i.d. from an unknown distribution. Contrastive SSL methods (Zbontar et al., 2021; Chen et al., 2020) learn an encoder $f$ that produces semantically meaningful representations by training on $\mathcal{D}_{\mathsf{real}}$. BarlowTwins, in particular, learns these representations using the cross-correlation matrix of the outputs of different augmented

views of a given batch of training data:

$$\mathcal{L}_{\text{BT}} := \sum_{i=1}^{d}(1 - F_{ii})^2 + \lambda \sum_{i=1}^{d} \sum_{j=1, j\neq i}^{d} F_{ij}^2, \tag{1}$$

where $F$ is the cross-correlation of *outputs within in a batch* $B$ s.t. $F_{ij} = \mathbb{E}_{x \in B}\mathbb{E}_{x_1, x_2 \in \mathcal{A}(x)}[f_i(x_1)f_j(x_2)]$ with $\mathcal{A}(x)$ being the set of augmented views of $x$, $d$ is the dimension of the encoder $f$ and $\lambda$ is a hyperparameter. After pre-training, a linear classifier is trained on representations and labels of downstream task(s).

Our goal is to generate a synthetic dataset $\mathcal{D}_{\text{syn}}$ from $\mathcal{D}_{\text{real}}$, such that $|\mathcal{D}_{\text{syn}}| \ll |\mathcal{D}_{\text{real}}|$ and the representations of $\mathcal{D}_{\text{real}}$ using the encoder $f_{\theta_{\text{syn}}}$, trained on the the synthetic data $\mathcal{D}_{\text{syn}}$, are similar to those obtained from encoder $f_{\theta_{\text{real}}}$, trained on the real data $\mathcal{D}_{\text{real}}$. Formally,

$$\mathcal{D}_{\text{syn}}^* := \underset{\mathcal{D}_{\text{syn}}}{\arg\min} \, \mathbb{E}_{x \sim \mathcal{D}_{\text{real}}} D(f_{\theta_{\text{syn}}}(x), f_{\theta_{\text{real}}}(x)), \tag{2}$$

where $D(\cdot, \cdot)$ is a distance function.

**Evaluation** To evaluate the encoder trained on the synthetic data, for every downstream task $\mathcal{D}_d$, we train a linear classifier $g_{\mathcal{D}_d}$ on the representations of $f_{\theta_{\text{syn}}}$, and evaluate the generalization error of the linear classifier on $\mathcal{D}_d$:

$$\text{Err}[f_{\theta_{\text{syn}}}(\mathcal{D}_d)] := \mathbb{E}_{(x,y) \sim \mathcal{D}_d} \mathbf{1}\big[g_{\mathcal{D}_d}(f_{\theta_{\text{syn}}}(x)) \neq y\big] \tag{3}$$

An effective encoder achieves small $\text{Err}[f_{\theta_{\text{syn}}}(\mathcal{D}_d)]$ across all downstream tasks.

## 4 MATCHING TRAINING TRAJECTORIES FOR SSL

As discussed in Sec. 2, distribution matching and gradient matching methods cannot work without labels, and meta-model matching cannot effectively update the encoder. Therefore, in our work, we focus on application of MTT to SSL distillation. First, we discuss the challenges of applying MTT in the SSL setting and show that its naïve application does not work. Then, we present our method, MKDT, that relies on recent results in knowledge distillation (KD) to enable trajectory matching for SSL.

### 4.1 CHALLENGES OF MATCHING SSL TRAINING TRAJECTORIES

In this section, we first introduce trajectory matching (MTT) for supervised learning (SL). Then, we discuss why naively applying MTT to the SSL setting does not work.

**Matching Training Trajectories for SL** MTT Cazenavette et al. (2022) generates a synthetic dataset by matching the trajectory of parameters $\hat{\theta}$ of a model trained on the synthetic data with trajectory of parameters $\theta^*$ of the model trained on real data (*expert trajectory*). This matching is guided by the following loss function:

$$\mathcal{L}_{DD}(\mathcal{D}_{\text{syn}}) = \frac{\|\hat{\theta}_{t+N} - \theta_{t+M}^*\|^2}{\|\theta_t^* - \theta_{t+M}^*\|^2} \tag{4}$$

In equation 4, $\theta_t^*$ denotes the model parameters after training on real data up to step $t$. The term $\hat{\theta}_{t+N}$ represents the model parameters after training on the synthetic dataset for $N$ steps, starting from $\theta_t^*$. Similarly, $\theta_{t+M}^*$ refers to the model parameters after $M$ steps of training on the real dataset. The primary goal of MTT is to ensure that the encoder's weights after training on the synthetic dataset for $N$ steps closely match the encoder's weights after training on real data for a significantly larger number of steps $M$, usually with $N \ll M$. MTT is agnostic to the training algorithm and doesn't rely on labels; thus, can be applied to dataset distillation for SSL. However, naïve application of MTT cannot effectively distill synthetic data for SSL pre-training, as we will discuss next.

**High Variance Gradients Prevent Effective Trajectory Matching for SSL** SSL losses rely on interaction between all examples in a batch and consequently have high variance over choices of random batches (c.f., the Barlow-Twins loss in equation 1). As a result, the contribution of examples to the loss and hence their gradients varies significantly based on the rest of examples in the batch Robinson et al. (2021), unlike SL where each example's contribution to the loss is independent of

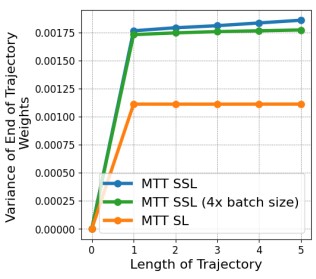
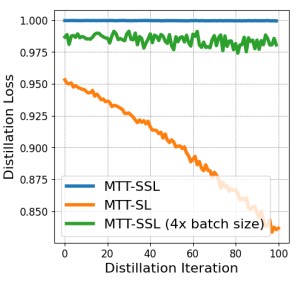
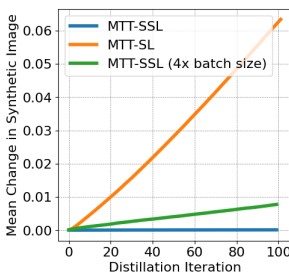

(a) Variance in Weights at End of Trajectory v/s Trajectory Length

(b) Distillation Loss (Error in Matching Trajectories) v/s Distillation Iterations

(c) Average Absolute Change in Pixel v/s # Distillation Iterations

Figure 1: Challenges of MTT for SSL (Dataset: CIFAR100 (1%); Arch: 3-layer ConvNet)

other examples. The high variance in gradient over mini-batches in each iteration results in high variance of the trajectories of SSL.

**Theoretical Evidence for Higher Variance Gradients in SSL.** We now present, in Theorem 4.1, theoretical evidence, in a simplified setting, demonstrating that the variance of the gradient of SSL over mini-batches is indeed greater than that of SL, i.e., $\text{Var}\big(\nabla_W L_{SSL}(B)\big) > \text{Var}\big(\nabla_W L_{SL}(B)\big)$. Proof appears in Appendix C. Appendix D presents a more general version of this analysis, when optimizing with synchronous parallel SGD.

**Theorem 4.1.** *Let $D = \{(x_i, y_i)\}_{i=1}^n$ be a dataset with $n$ examples, where $x_i$ is the $i$-th input and $y_i \in \{0, 1\}$ is its corresponding class label. Assume the data $x_i$ are generated using the sparse coding model Xue et al. (2023); Joshi et al. (2024): for class 0, $x_i = e_0 + \epsilon_i$, and for class 1, $x_i = e_1 + \epsilon_i$, where $e_0$ and $e_1$ are basis vectors and $\epsilon_i \sim \mathcal{N}(0, \sigma_N I)$ is noise. Note that using mean class vectors $e_0, e_1$ w.l.o.g. models the setting of arbitrary mean class vectors that are orthogonal to each other. Each class has $\frac{n}{2}$ examples.*

*Consider a linear model $f_\theta(x) = Wx$, with $W$ initialized as $I$ (the identity matrix). The supervised mean squared error (MSE) loss is given by:*

$$L_{SL}(B) = \frac{1}{|B|} \sum_{i \in B} \|f_\theta(x_i) - e_{y_i}\|^2,$$

*where $e_{y_i}$ is the one-hot encoded vector for class $y_i$, and $B$ is a mini-batch.*

*The SSL Loss (spectral contrastive loss used here for simplicity of analysis) is given by:*

$$L_{SSL} = -2\mathbb{E}_{\substack{x_1, x_2 \sim \mathcal{A}(x_i) \\ x_i \in B}} \left[ f_\theta(x_1)^T f_\theta(x_2) \right] + \mathbb{E}_{x_i, x_j \in B} \left[ f_\theta(x_i)^T f_\theta(x_j) \right]^2,$$

*where $\mathcal{A}(x_i) = x_i + \epsilon_{aug}$, with $\epsilon_{aug} \sim \mathcal{N}(0, I)$ representing augmentation noise.*

*Under stochastic gradient descent (SGD) with a mini-batch size B of 2:*

$$\text{Var}(\nabla_W L_{SL}(B)) < \text{Var}(\nabla_W L_{SSL}(B)).$$

*where for a matrix M, $\text{Var}(M) := \mathbb{E}[\|M - \mathbb{E}[M]\|^2]$ as in Gower et al. (2020).*

**Empirical Evidence for Challenges of Matching SSL Trajectories** Due to the high variance in the gradient of SSL objectives, the naïve application of MTT to SSL does not succeed. Firstly, the slower convergence caused by high variance gradients necessitates much longer trajectories for both training on real and synthetic data. Secondly, the higher variance of gradients results in greater variance in the weights at the end of trajectories starting from the same initialization (henceforth referred to as *variance of trajectories*), as illustrated theoretically in the simplified setting above. Attempting to match SSL's longer, higher variance trajectories is extremely challenging, as matching such trajectories results in chaotic updates to the synthetic images. Thus, the synthetic images cannot move away from their initialization meaningfully. Fig. 1a shows empirically that the variance of SSL trajectories is larger than that of SL trajectories, across different trajectory lengths. Additionally, the

variance of trajectories grows faster, with length of trajectory, for SSL than for SL, exacerbating the problem for longer trajectory matching. Fig. 1b compares a simplified distillation using MTT with a single expert trajectory for SSL and SL. Despite extensive hyper-parameter tuning, matching even a single expert trajectory is challenging for SSL, confirmed by the slow decrease of distillation loss. This indicates that the training trajectory on the distilled set is unable to match the training trajectory on the real data for SSL. Fig. 1c shows that the difficulty in aligning trajectories is due to the chaotic updates of the synthetic image, as evidenced by the synthetic images being unable to move away from their initialization. To further confirm that the inability to distill effectively is indeed due to the variance of trajectories, we also include a comparison to MTT SSL with 4x larger batch size, which leads to slightly lower variance. Fig. 1a confirms that indeed the larger batch size reduces the variance of the trajectories slightly. However, Fig. 1b and 1c show that reducing the variance of SSL trajectories via larger batch size is insufficient to help distillation since an infeasibly large batch size will likely be required to achieve the necessary low variance trajectories.

Next, we will present our method, MKDT, designed to address the above challenges.

## 4.2 Matching Knowledge Distillation Trajectories

To reduce the length and variance of SSL trajectories, our key idea is to leverage the recent results in knowledge distillation (KD) Kim et al. (2021). We first introduce KD, and then discuss our method, MKDT, Matching Knowledge Distillation Trajectories, that leverages KD to reduce the length and variance of SSL trajectories.

**Knowledge Distillation (KD)** KD refers to *distilling* the *knowledge* of a larger model (teacher) into a smaller model (student) to achieve similar generalization as the larger model, but with reduced complexity and faster inference times. Here, we rely on the knowledge distillation objective for SSL models, introduced in Yu et al. (2019):

$$\mathcal{L}_{\mathcal{KD}} = \mathbb{E}_{x \sim \mathcal{D}_{\mathsf{real}}}\big[\mathrm{MSE}(f_S(x), f_T(x))\big], \tag{5}$$

where $f_S$ and $f_T$ represent the student and teacher encoders respectively. (Yu et al., 2019) trains student models with the aforementioned KD objective and the original SSL Loss. However, we only minimize the MSE between student and teacher representations to avoid the issues with matching SSL training trajectories (discussed in Sec. 4.1).

**Converting SSL to SL trajectories via KD** We use the objective presented in equation 5 i.e. minimizing the MSE between the *representations* of a student and a teacher model trained with SSL. In doing so, we train the student model to match the performance of the teacher trained with SSL. Note that training the student model by minimizing the MSE loss in equation 5 is a *supervised* objective. Therefore, while the trained student model will produce similar representations to that of the teacher, training with MSE loss is much faster than SSL, as its gradients have a much smaller variance (*c.f.* Fig. 1a). Thus, we can get shorter and lower variance *expert* trajectories from the *student* models trained with KD using the MSE loss, instead of the teacher model trained with SSL. Then, we can generate synthetic examples by matching these shorter and lower variance trajectories, without relying on labels.

**Matching KD Trajectories (MKDT)** We now describe our method MKDT: Matching Knowledge Distillation Trajectories. MKDT has the following three steps: (1) training a teacher model with SSL, (2) getting expert trajectories by training student models to match representations of the teacher using KD objective, (3) generating synthetic examples by matching the expert trajectories. Below, we discuss each of these steps in more details.

**(1) Training the Teacher Model with SSL** First, we train the teacher encoder $f_{\theta_T}$ with $\mathcal{L}_{\mathsf{SSL}}$ on $\mathcal{D}_{\mathsf{real}}$:

$$\theta_T = \arg\min_\theta \big[\mathcal{L}_{\mathsf{SSL}}(f_\theta, \mathcal{D}_{\mathsf{real}})\big], \tag{6}$$

In our experiments, $\mathcal{L}_{\mathsf{SSL}}$ is the BarlowTwins loss function shown in equation 1, but our method is agnostic to the choice of SSL algorithm. Since KD with larger models leads to better downstream performance (Huang et al., 2022), we use a teacher model that is much larger than the student encoder used for creating the expert trajectories for distillation. For example, in our experiments we use a ResNet-18 as the teacher encoder and a 3 or 4-layer ConvNet as the student encoder.

**(2) Getting Expert Trajectories with KD** For training the expert trajectories, we encode the full real data with the teacher model and train the student model to minimize the MSE between its

---

**Algorithm 1** MKDT: Matching Knowledge Distillation Trajectories

---

**Require:** $K$: Number of expert trajectories
**Require:** $S$: Number of distillation steps
**Require:** $M$: # of updates between starting and target expert params.
**Require:** $N$: # of updates to student network per distillation step.
**Require:** $T^+ < T$: Maximum start epoch.
 1: Train model $f_{\theta_T}$ using $\mathcal{L}_{\mathsf{SSL}}$ on $\mathcal{D}_{\mathsf{real}}$ using augmentations $\mathcal{A}$
 2: $\{\tau_i^*\} \leftarrow$ Train $K$ expert trajectories to minimize $\mathcal{L}_{\mathcal{KD}}(f_{s_i}, f_{\theta_T})$
 3: Initialize distilled data $\mathcal{D}_{\mathsf{syn}}, \mathcal{Z}_{\mathsf{syn}} \sim \mathcal{D}_{\mathsf{real}}, \mathcal{Z}_{\mathsf{real}}$
 4: Initialize trainable learning rate $\alpha_{\mathsf{syn}} \coloneqq \alpha_0$ for $\mathcal{D}_{\mathsf{syn}}$
 5: **for** $S$ steps **do**
 6:   ▷ Sample expert trajectory: $\tau^* \sim \{\tau_i^*\}$ with $\tau^* = \{\theta_t^*\}_0^T$
 7:   ▷ Choose random start epoch, $t \leq T^+$
 8:   ▷ Initialize student network with expert params:
 9:     $\hat{\theta}_t \coloneqq \theta_t^*$
10:   **for** $n = 0 \rightarrow N - 1$ **do**
11:     ▷ Sample a mini-batch of distilled images:
12:       $b_{t+n} \sim \mathcal{D}_{\mathsf{syn}}$
13:     ▷ Update student network w.r.t. MSE loss to reference representations:
14:     $\hat{\theta}_{t+n+1} = \hat{\theta}_{t+n} - \alpha_{\mathsf{syn}} \nabla \mathcal{L}_{\mathsf{MSE}_{\hat{\theta}_{t+n}}}(b_{t+n}, \mathcal{Z}_{\mathsf{syn}})$
15:   **end for**
16:   ▷ Compute loss between ending student and expert params:
17:     $\mathcal{L}_{DD}(\mathcal{D}_{\mathsf{syn}}) = \|\hat{\theta}_{t+N} - \theta_{t+M}^*\|_2^2 \,/\, \|\theta_t^* - \theta_{t+M}^*\|_2^2$
18:   ▷ Update $\mathcal{D}_{\mathsf{syn}}$ and $\alpha_{\mathsf{syn}}$ with respect to $\mathcal{L}_{DD}(\mathcal{D}_{\mathsf{syn}})$
19: **end for**
**Ensure:** distilled data $\mathcal{D}_{\mathsf{syn}}, \mathcal{Z}_{\mathsf{syn}}$ and learning rate $\alpha_{\mathsf{syn}}$

---

representations and that of the teacher model. We refer to the representations of the real data from the teacher model as *teacher representations* denoted by $\mathcal{Z}_T = \left[ \cdots f_{\theta_T}(x_i) \cdots \right], \forall x_i \in \mathcal{D}_{\mathsf{real}}$. Formally,

$$\min_{\theta^*} \mathbb{E}_{x_i \in \mathcal{D}_{\mathsf{real}}} \mathcal{L}_{\mathsf{MSE}}(f_{\theta^*}(x_i), [\mathcal{Z}_T]_i). \tag{7}$$

We train $K$ such student encoders and save the weights after each epoch of training to generate the *expert* trajectories that we will match in the distillation phase.

**(3) Data Distillation by Matching KD Trajectories** We now optimize the synthetic data such that training on it results in trajectories that match the aforementioned expert trajectories. First, we initialize $\mathcal{D}_{\mathsf{syn}}$ as a subset of $\mathcal{D}_{\mathsf{real}}$ and $\mathcal{Z}_{\mathsf{syn}}$ as the corresponding teacher representations from $\mathcal{Z}_T$. Then, in every distillation iteration, we sample an expert trajectory starting at epoch $t$, where $t \leq T^+$, of length $M$. We then train on the synthetic data for $N$ steps by minimizing the MSE between representations of synthetic data from $f_{\theta_{\mathsf{syn}}}$ and $\mathcal{Z}_{\mathsf{syn}}$ $f_{\theta_T}$. Formally, $\forall n \in [N]$,

$$\hat{\theta}_{t+n+1} = \hat{\theta}_{t+n} - \alpha_{\mathsf{syn}} \nabla \mathcal{L}_{\mathsf{MSE}}(f_{\hat{\theta}_{t+n}}(\mathcal{D}_{\mathsf{syn}}), \mathcal{Z}_{\mathsf{syn}}) \tag{8}$$

Now, we compute our distillation loss $\mathcal{L}_{DD}(\mathcal{D}_{\mathsf{syn}})$ (shown in equation 4) using the parameters of the encoder trained on the synthetic data and the encoder trained on the full data, and update the synthetic data and learning rate, $\mathcal{D}_{\mathsf{syn}}$ and $\alpha_{\mathsf{syn}}$, respectively, to minimize this. Note that $\mathcal{Z}_{\mathsf{syn}}$ remains unchanged. We repeat this distillation for $S$ iterations. Pseudo-code of MKDT is provided in Alg. 1.

**Initializing Synthetic Data** Empirically, we find that initializing $\mathcal{D}_{\mathsf{syn}}$ from the subset of examples from $\mathcal{D}_{\mathsf{real}}$ that have *high loss* across the expert trajectories, leads to better downstream performance than initializing with random subsets. In particular, for all expert trajectories, we use the encoders after 1 epoch of training and use it to compute the MSE loss for all examples $x_i \in \mathcal{D}_{\mathsf{real}}$ i.e. $\mathcal{L}_{\mathsf{MSE}}(f_{\theta_1^*}(x_i), [\mathcal{Z}_T]_i)$. We then average the loss for examples across encoders from all expert trajectories and choose the examples with highest loss to initialize our synthetic data. Sec. 5 compares initializing MKDT with random subsets and such *high loss* subsets.

**Evaluating the Distilled Dataset** For evaluation, we first pre-train the encoder $f_{\theta_{\mathsf{syn}}}$ on the distilled data by minimizing the MSE between the representations of the synthetic data $\mathcal{D}_{\mathsf{syn}}$ and $\mathcal{Z}_{\mathsf{syn}}$ using the distilled learning rate $\alpha_{\mathsf{syn}}$.

$$\theta_{\mathsf{syn}} = \arg\min_{\theta} \mathbb{E}_{x_i \in \mathcal{D}_{\mathsf{syn}}} \mathcal{L}_{\mathsf{MSE}}(f_\theta(x), [\mathcal{Z}_{\mathsf{syn}}]_i) \text{ with l.r. } \alpha_{\mathsf{syn}} \tag{9}$$

Table 1: Pre-training on CIFAR10 (2% of Full Data)

| Size of Downstream Labeled Data (%) | Method | Pre-Training | Downstream | | | | | |
|---|---|---|---|---|---|---|---|---|
| | | CIFAR10 | Tiny ImageNet | CIFAR100 | Aircraft | CUB2011 | Dogs | Flowers |
| 1% | No Pre-Training | $35.84_{\pm1.39}$ | $2.52_{\pm0.09}$ | $8.01_{\pm0.19}$ | $2.43_{\pm0.17}$ | $1.27_{\pm0.10}$ | $1.92_{\pm0.18}$ | $2.02_{\pm0.76}$ |
| | Random Subset | $36.35_{\pm0.67}$ | $2.41_{\pm0.08}$ | $7.42_{\pm0.31}$ | $2.41_{\pm0.30}$ | $1.16_{\pm0.08}$ | $1.90_{\pm0.22}$ | $1.99_{\pm0.19}$ |
| | SAS Subset | $36.02_{\pm1.52}$ | $2.69_{\pm0.31}$ | $7.77_{\pm0.35}$ | $2.29_{\pm0.26}$ | $1.14_{\pm0.01}$ | $1.78_{\pm0.32}$ | $2.22_{\pm0.23}$ |
| | KRR-ST | $37.19_{\pm0.49}$ | $2.84_{\pm0.17}$ | $8.67_{\pm0.41}$ | $2.53_{\pm0.05}$ | $1.25_{\pm0.02}$ | $1.88_{\pm0.32}$ | $2.42_{\pm0.22}$ |
| | **MKDT** | $\mathbf{44.36}_{\pm1.61}$ | $\mathbf{3.58}_{\pm0.09}$ | $\mathbf{10.58}_{\pm0.24}$ | $\mathbf{2.58}_{\pm0.18}$ | $\mathbf{1.37}_{\pm0.09}$ | $\mathbf{2.11}_{\pm0.28}$ | $\mathbf{2.52}_{\pm0.07}$ |
| | Full Data | $58.21_{\pm0.28}$ | $4.94_{\pm0.38}$ | $14.53_{\pm0.40}$ | $2.92_{\pm0.25}$ | $1.69_{\pm0.13}$ | $2.40_{\pm0.25}$ | $3.23_{\pm0.69}$ |
| 5% | No Pre-Training | $46.23_{\pm0.07}$ | $5.37_{\pm0.39}$ | $16.12_{\pm0.13}$ | $5.61_{\pm0.68}$ | $1.97_{\pm0.13}$ | $2.90_{\pm0.18}$ | $\mathbf{5.22}_{\pm0.52}$ |
| | Random Subset | $46.62_{\pm1.02}$ | $5.49_{\pm0.12}$ | $15.28_{\pm0.66}$ | $5.35_{\pm0.96}$ | $1.51_{\pm0.08}$ | $2.52_{\pm0.22}$ | $3.64_{\pm0.46}$ |
| | SAS Subset | $46.52_{\pm0.61}$ | $5.41_{\pm0.42}$ | $15.90_{\pm0.28}$ | $5.63_{\pm0.76}$ | $1.48_{\pm0.16}$ | $2.69_{\pm0.21}$ | $3.75_{\pm0.10}$ |
| | KRR-ST | $46.75_{\pm0.45}$ | $6.85_{\pm0.20}$ | $16.65_{\pm0.31}$ | $5.41_{\pm0.45}$ | $1.88_{\pm0.10}$ | $2.76_{\pm0.40}$ | $4.52_{\pm0.14}$ |
| | **MKDT** | $\mathbf{53.08}_{\pm0.13}$ | $\mathbf{7.25}_{\pm0.17}$ | $\mathbf{19.57}_{\pm0.29}$ | $\mathbf{5.97}_{\pm0.79}$ | $\mathbf{2.06}_{\pm0.10}$ | $\mathbf{3.06}_{\pm0.46}$ | $4.97_{\pm0.54}$ |
| | Full Data | $67.16_{\pm0.43}$ | $10.85_{\pm0.16}$ | $26.38_{\pm0.52}$ | $6.92_{\pm0.61}$ | $2.51_{\pm0.08}$ | $3.88_{\pm0.25}$ | $6.37_{\pm0.67}$ |

We then evaluate the encoder $f_{\theta_{\mathsf{syn}}}$ using $\mathrm{Err}_{f_{\theta_{\mathsf{syn}}}}(\mathcal{D}_d)$, defined in equation 3, i.e. the generalization error of linear classifier $g_{\mathcal{D}_d}$ trained on the representations obtained from encoder $f_{\theta_{\mathsf{syn}}}$ and corresponding labels of downstream task $\mathcal{D}_d$.

## 5 EXPERIMENTS

Table 2: Pre-training on CIFAR100 (2% of Full Data)

| Size of Downstream Labeled Data (%) | Method | Pre-Training | Downstream | | | | | |
|---|---|---|---|---|---|---|---|---|
| | | CIFAR100 | Tiny ImageNet | CIFAR10 | Aircraft | CUB2011 | Dogs | Flowers |
| 1% | No Pre-Training | $8.01_{\pm0.19}$ | $2.52_{\pm0.09}$ | $35.84_{\pm1.39}$ | $2.43_{\pm0.17}$ | $1.27_{\pm0.10}$ | $1.92_{\pm0.18}$ | $2.02_{\pm0.76}$ |
| | Random Subset | $9.20_{\pm0.15}$ | $3.16_{\pm0.21}$ | $38.03_{\pm1.22}$ | $2.41_{\pm0.15}$ | $1.43_{\pm0.12}$ | $1.99_{\pm0.10}$ | $2.81_{\pm0.43}$ |
| | SAS Subset | $9.39_{\pm0.18}$ | $3.23_{\pm0.19}$ | $38.73_{\pm1.48}$ | $2.53_{\pm0.04}$ | $1.42_{\pm0.04}$ | $2.07_{\pm0.14}$ | $2.95_{\pm0.37}$ |
| | High Loss Subset | $10.03_{\pm0.12}$ | $3.33_{\pm0.10}$ | $39.78_{\pm1.61}$ | $2.47_{\pm0.29}$ | $1.56_{\pm0.14}$ | $2.13_{\pm0.24}$ | $2.63_{\pm0.51}$ |
| | KRR-ST | $8.31_{\pm0.30}$ | $2.73_{\pm0.08}$ | $37.19_{\pm0.83}$ | $2.56_{\pm0.20}$ | $1.29_{\pm0.04}$ | $1.92_{\pm0.11}$ | $2.58_{\pm0.14}$ |
| | **MKDT (Rnd Sub)** | $11.44_{\pm0.36}$ | $3.90_{\pm0.20}$ | $43.35_{\pm1.08}$ | $2.53_{\pm0.22}$ | $\mathbf{1.66}_{\pm0.13}$ | $\mathbf{2.22}_{\pm0.20}$ | $2.63_{\pm1.02}$ |
| | **MKDT** | $\mathbf{12.36}_{\pm0.67}$ | $\mathbf{4.13}_{\pm0.29}$ | $\mathbf{44.90}_{\pm1.18}$ | $\mathbf{2.74}_{\pm0.30}$ | $1.61_{\pm0.14}$ | $2.15_{\pm0.39}$ | $\mathbf{3.24}_{\pm0.44}$ |
| | Full Data | $21.44_{\pm0.86}$ | $6.80_{\pm0.37}$ | $58.21_{\pm0.81}$ | $3.20_{\pm0.22}$ | $1.79_{\pm0.08}$ | $2.50_{\pm0.27}$ | $3.09_{\pm1.14}$ |
| 5% | No Pre-Training | $16.12_{\pm0.13}$ | $5.37_{\pm0.39}$ | $46.23_{\pm0.07}$ | $5.61_{\pm0.68}$ | $1.97_{\pm0.13}$ | $2.90_{\pm0.18}$ | $5.22_{\pm0.52}$ |
| | Random Subset | $17.75_{\pm0.42}$ | $6.79_{\pm0.06}$ | $48.59_{\pm0.26}$ | $5.66_{\pm0.71}$ | $2.12_{\pm0.23}$ | $3.02_{\pm0.31}$ | $5.44_{\pm0.22}$ |
| | SAS Subset | $17.94_{\pm0.54}$ | $6.71_{\pm0.52}$ | $48.69_{\pm0.26}$ | $5.95_{\pm0.88}$ | $2.15_{\pm0.27}$ | $3.22_{\pm0.47}$ | $5.56_{\pm0.43}$ |
| | High Loss Subset | $18.72_{\pm0.21}$ | $6.94_{\pm0.34}$ | $49.59_{\pm0.34}$ | $5.63_{\pm0.52}$ | $2.58_{\pm0.13}$ | $3.18_{\pm0.20}$ | $6.14_{\pm0.54}$ |
| | KRR-ST | $16.40_{\pm0.63}$ | $6.16_{\pm0.45}$ | $47.96_{\pm0.32}$ | $5.54_{\pm0.97}$ | $2.00_{\pm0.08}$ | $2.95_{\pm0.25}$ | $4.69_{\pm0.15}$ |
| | **MKDT (Rnd Sub)** | $21.71_{\pm0.28}$ | $8.01_{\pm0.08}$ | $53.08_{\pm0.19}$ | $6.24_{\pm0.79}$ | $\mathbf{2.53}_{\pm0.03}$ | $\mathbf{3.38}_{\pm0.23}$ | $6.26_{\pm0.22}$ |
| | **MKDT** | $\mathbf{22.64}_{\pm0.42}$ | $\mathbf{8.07}_{\pm0.16}$ | $\mathbf{54.12}_{\pm0.29}$ | $\mathbf{6.68}_{\pm0.83}$ | $2.50_{\pm0.16}$ | $3.25_{\pm0.36}$ | $\mathbf{6.37}_{\pm0.46}$ |
| | Full Data | $35.78_{\pm0.54}$ | $14.11_{\pm0.55}$ | $67.25_{\pm0.49}$ | $7.46_{\pm0.53}$ | $3.00_{\pm0.05}$ | $4.23_{\pm0.21}$ | $8.41_{\pm0.60}$ |

Table 3: Pre-training on TinyImageNet (2% of Full Data)

| Size of Downstream Labeled Data (%) | Method | Pre-Training | Downstream | | | | | |
|---|---|---|---|---|---|---|---|---|
| | | Tiny ImageNet | CIFAR10 | CIFAR100 | Aircraft | CUB2011 | Dogs | Flowers |
| 1% | No Pre-Training | $2.63_{\pm0.22}$ | $30.40_{\pm1.02}$ | $7.14_{\pm0.35}$ | $2.25_{\pm0.11}$ | $1.28_{\pm0.16}$ | $1.76_{\pm0.13}$ | $2.84_{\pm0.20}$ |
| | Random Subset | $3.03_{\pm0.20}$ | $34.46_{\pm1.13}$ | $7.66_{\pm0.67}$ | $2.27_{\pm0.22}$ | $1.24_{\pm0.05}$ | $1.92_{\pm0.26}$ | $2.16_{\pm0.47}$ |
| | KRR-ST | $3.32_{\pm0.22}$ | $34.24_{\pm0.94}$ | $7.84_{\pm0.98}$ | $2.14_{\pm0.30}$ | $1.30_{\pm0.08}$ | $1.95_{\pm0.22}$ | $2.21_{\pm0.33}$ |
| | **MKDT** | $\mathbf{3.87}_{\pm0.05}$ | $\mathbf{37.25}_{\pm0.47}$ | $\mathbf{8.95}_{\pm0.34}$ | $\mathbf{2.30}_{\pm0.19}$ | $\mathbf{1.36}_{\pm0.22}$ | $\mathbf{1.99}_{\pm0.20}$ | $\mathbf{2.86}_{\pm0.25}$ |
| | Full Data | $9.42_{\pm0.36}$ | $50.52_{\pm0.87}$ | $15.18_{\pm0.37}$ | $2.57_{\pm0.29}$ | $1.59_{\pm0.31}$ | $2.20_{\pm0.37}$ | $3.29_{\pm0.28}$ |
| 5% | No Pre-Training | $5.69_{\pm0.45}$ | $39.91_{\pm0.36}$ | $13.32_{\pm0.30}$ | $4.46_{\pm0.81}$ | $1.71_{\pm0.06}$ | $2.51_{\pm0.18}$ | $4.83_{\pm0.32}$ |
| | Random Subset | $6.76_{\pm0.16}$ | $43.74_{\pm0.82}$ | $13.83_{\pm0.13}$ | $4.49_{\pm0.91}$ | $1.66_{\pm0.14}$ | $2.67_{\pm0.31}$ | $4.23_{\pm0.70}$ |
| | KRR-ST | $7.13_{\pm0.22}$ | $42.44_{\pm1.85}$ | $13.85_{\pm0.72}$ | $3.99_{\pm0.57}$ | $1.77_{\pm0.07}$ | $2.47_{\pm0.21}$ | $4.14_{\pm0.71}$ |
| | **MKDT** | $\mathbf{7.99}_{\pm0.32}$ | $\mathbf{45.97}_{\pm0.27}$ | $\mathbf{16.50}_{\pm0.35}$ | $\mathbf{4.66}_{\pm0.70}$ | $\mathbf{2.07}_{\pm0.11}$ | $\mathbf{2.91}_{\pm0.10}$ | $\mathbf{5.49}_{\pm0.51}$ |
| | Full Data | $18.93_{\pm0.34}$ | $58.90_{\pm0.43}$ | $26.47_{\pm0.78}$ | $5.07_{\pm0.71}$ | $2.47_{\pm0.06}$ | $3.85_{\pm0.19}$ | $7.09_{\pm1.01}$ |

In this section, we evaluate the downstream generalization of models trained using the synthetic sets distilled by MKDT that are 2% and 5% of the original dataset's size for CIFAR10, CIFAR100 (Krizhevsky & Hinton, 2009), TinyImageNet (Le & Yang, 2015). We also conduct ablation studies over initialization of the distilled set and the SSL algorithm. Finally, we also consider the generalization of the distilled sets to larger architectures.

Table 4: Pre-training with Larger Distilled Set Size (5% of Full Data)

| Pre-Training Dataset | Size of Downstream Labeled Data (%) | Method | Downstream Task Accuracy | | | | | | |
|---|---|---|---|---|---|---|---|---|---|
| | | | CIFAR10 | CIFAR100 | Tiny ImageNet | Aircraft | CUB2011 | Dogs | Flowers |
| CIFAR10 | 1% | Random Subset | $37.63_{\pm 2.28}$ | $7.63_{\pm 0.13}$ | $2.63_{\pm 0.07}$ | $2.28_{\pm 0.22}$ | $1.13_{\pm 0.11}$ | $1.81_{\pm 0.18}$ | $2.05_{\pm 0.23}$ |
| | | KRR-ST | $36.69_{\pm 0.88}$ | $8.69_{\pm 0.32}$ | $3.20_{\pm 0.23}$ | $2.26_{\pm 0.13}$ | $1.33_{\pm 0.09}$ | $1.91_{\pm 0.34}$ | $2.39_{\pm 0.18}$ |
| | | **MKDT** | $\mathbf{50.23}_{\pm 1.48}$ | $\mathbf{12.33}_{\pm 0.31}$ | $\mathbf{4.27}_{\pm 0.36}$ | $\mathbf{3.11}_{\pm 0.15}$ | $\mathbf{1.59}_{\pm 0.07}$ | $\mathbf{2.26}_{\pm 0.29}$ | $\mathbf{2.42}_{\pm 0.62}$ |
| | 5% | Random Subset | $48.13_{\pm 0.35}$ | $16.06_{\pm 0.15}$ | $5.21_{\pm 0.56}$ | $5.03_{\pm 0.88}$ | $1.71_{\pm 0.12}$ | $2.61_{\pm 0.20}$ | $3.30_{\pm 0.10}$ |
| | | KRR-ST | $47.40_{\pm 0.34}$ | $16.95_{\pm 0.53}$ | $7.10_{\pm 0.27}$ | $5.56_{\pm 0.77}$ | $1.98_{\pm 0.07}$ | $2.78_{\pm 0.16}$ | $4.38_{\pm 0.04}$ |
| | | **MKDT** | $\mathbf{58.37}_{\pm 0.17}$ | $\mathbf{23.15}_{\pm 0.71}$ | $\mathbf{8.84}_{\pm 0.20}$ | $\mathbf{6.79}_{\pm 0.88}$ | $\mathbf{2.30}_{\pm 0.14}$ | $\mathbf{3.34}_{\pm 0.35}$ | $\mathbf{5.94}_{\pm 0.24}$ |
| CIFAR100 | 1% | Random Subset | $42.45_{\pm 0.76}$ | $11.55_{\pm 0.37}$ | $4.17_{\pm 0.15}$ | $2.47_{\pm 0.17}$ | $1.51_{\pm 0.12}$ | $2.12_{\pm 0.18}$ | $2.61_{\pm 0.61}$ |
| | | KRR-ST | $37.86_{\pm 1.14}$ | $9.02_{\pm 0.24}$ | $2.94_{\pm 0.13}$ | $2.42_{\pm 0.35}$ | $1.50_{\pm 0.07}$ | $1.99_{\pm 0.19}$ | $\mathbf{3.04}_{\pm 0.36}$ |
| | | **MKDT** | $\mathbf{47.77}_{\pm 1.12}$ | $\mathbf{13.40}_{\pm 0.31}$ | $\mathbf{4.45}_{\pm 0.33}$ | $\mathbf{2.93}_{\pm 0.42}$ | $\mathbf{1.61}_{\pm 0.02}$ | $\mathbf{2.22}_{\pm 0.27}$ | $2.59_{\pm 1.07}$ |
| | 5% | Random Subset | $51.92_{\pm 0.33}$ | $20.34_{\pm 0.20}$ | $7.83_{\pm 0.24}$ | $6.06_{\pm 0.79}$ | $2.27_{\pm 0.18}$ | $3.22_{\pm 0.39}$ | $5.78_{\pm 0.34}$ |
| | | KRR-ST | $47.53_{\pm 0.11}$ | $17.24_{\pm 0.47}$ | $6.60_{\pm 0.32}$ | $5.37_{\pm 0.85}$ | $2.31_{\pm 0.33}$ | $2.87_{\pm 0.27}$ | $5.23_{\pm 0.14}$ |
| | | **MKDT** | $\mathbf{56.61}_{\pm 0.58}$ | $\mathbf{25.18}_{\pm 0.67}$ | $\mathbf{9.12}_{\pm 0.40}$ | $\mathbf{6.66}_{\pm 0.62}$ | $\mathbf{2.66}_{\pm 0.23}$ | $\mathbf{3.66}_{\pm 0.44}$ | $\mathbf{6.93}_{\pm 0.60}$ |
| TinyImageNet | 1% | Random Subset | $40.33_{\pm 1.16}$ | $9.41_{\pm 0.29}$ | $4.19_{\pm 0.19}$ | $\mathbf{2.23}_{\pm 0.38}$ | $\mathbf{1.44}_{\pm 0.10}$ | $\mathbf{2.06}_{\pm 0.12}$ | $2.77_{\pm 0.24}$ |
| | | KRR-ST | $34.27_{\pm 1.36}$ | $7.54_{\pm 0.35}$ | $3.19_{\pm 0.22}$ | $2.11_{\pm 0.23}$ | $1.30_{\pm 0.12}$ | $1.68_{\pm 0.20}$ | $2.65_{\pm 0.64}$ |
| | | **MKDT** | $\mathbf{41.44}_{\pm 0.85}$ | $\mathbf{10.29}_{\pm 0.38}$ | $\mathbf{5.09}_{\pm 0.45}$ | $2.16_{\pm 0.28}$ | $1.29_{\pm 0.06}$ | $2.02_{\pm 0.28}$ | $\mathbf{2.92}_{\pm 0.49}$ |
| | 5% | Random Subset | $48.46_{\pm 0.40}$ | $15.63_{\pm 0.62}$ | $8.99_{\pm 0.61}$ | $4.55_{\pm 0.80}$ | $1.98_{\pm 0.18}$ | $\mathbf{2.91}_{\pm 0.47}$ | $5.06_{\pm 0.84}$ |
| | | KRR-ST | $42.82_{\pm 0.46}$ | $13.71_{\pm 0.30}$ | $6.50_{\pm 0.23}$ | $4.36_{\pm 0.49}$ | $1.97_{\pm 0.06}$ | $2.75_{\pm 0.37}$ | $3.97_{\pm 0.14}$ |
| | | **MKDT** | $\mathbf{50.79}_{\pm 0.47}$ | $\mathbf{19.25}_{\pm 0.23}$ | $\mathbf{10.63}_{\pm 0.23}$ | $\mathbf{4.88}_{\pm 0.65}$ | $\mathbf{2.08}_{\pm 0.03}$ | $2.89_{\pm 0.41}$ | $\mathbf{5.58}_{\pm 0.43}$ |

Table 5: Transfer to Larger Architectures (Pre-Training on CIFAR100 5%, 5% Downsteam Labels)

| | Method | Pre-Training | Downstream | | | | | |
|---|---|---|---|---|---|---|---|---|
| | | CIFAR100 | CIFAR10 | TinyImageNet | Aircraft | CUB2011 | Dogs | Flowers |
| ResNet-10 | No Pre-Training | $1.36_{\pm 0.31}$ | $13.18_{\pm 1.74}$ | $1.03_{\pm 0.07}$ | $1.00_{\pm 0.01}$ | $0.43_{\pm 0.13}$ | $0.60_{\pm 0.01}$ | $0.75_{\pm 0.14}$ |
| | Random | $18.80_{\pm 0.58}$ | $44.24_{\pm 0.85}$ | $10.33_{\pm 0.16}$ | $\mathbf{2.15}_{\pm 0.34}$ | $\mathbf{1.81}_{\pm 0.16}$ | $2.52_{\pm 0.29}$ | $5.41_{\pm 0.61}$ |
| | KRR-ST | $13.84_{\pm 0.78}$ | $39.21_{\pm 0.55}$ | $8.04_{\pm 0.52}$ | $2.12_{\pm 0.15}$ | $1.16_{\pm 0.05}$ | $1.77_{\pm 0.14}$ | $4.56_{\pm 0.42}$ |
| | **MKDT** | $\mathbf{23.23}_{\pm 0.79}$ | $\mathbf{49.13}_{\pm 0.69}$ | $\mathbf{13.35}_{\pm 0.24}$ | $1.68_{\pm 0.10}$ | $1.67_{\pm 0.09}$ | $\mathbf{2.64}_{\pm 0.16}$ | $\mathbf{6.15}_{\pm 0.47}$ |
| ResNet-18 | No Pre-Training | $1.01_{\pm 0.01}$ | $10.00_{\pm 0.00}$ | $0.91_{\pm 0.10}$ | $1.01_{\pm 0.01}$ | $0.56_{\pm 0.07}$ | $0.67_{\pm 0.09}$ | $0.93_{\pm 0.38}$ |
| | Random | $16.82_{\pm 0.69}$ | $40.11_{\pm 1.16}$ | $8.95_{\pm 0.23}$ | $1.84_{\pm 0.25}$ | $1.62_{\pm 0.06}$ | $\mathbf{2.40}_{\pm 0.25}$ | $5.16_{\pm 0.59}$ |
| | KRR-ST | $12.30_{\pm 0.83}$ | $35.73_{\pm 1.07}$ | $7.21_{\pm 0.35}$ | $\mathbf{2.32}_{\pm 0.39}$ | $1.18_{\pm 0.16}$ | $1.81_{\pm 0.14}$ | $2.45_{\pm 0.12}$ |
| | **MKDT** | $\mathbf{21.51}_{\pm 0.17}$ | $\mathbf{46.10}_{\pm 0.60}$ | $\mathbf{11.57}_{\pm 0.17}$ | $2.05_{\pm 0.43}$ | $\mathbf{1.86}_{\pm 0.05}$ | $2.36_{\pm 0.29}$ | $\mathbf{5.17}_{\pm 0.93}$ |

**Distillation Setup** We use ResNet-18 trained with Barlow Twins (Zbontar et al., 2021) as the teacher encoder and train $K = 100$ student encoders (using ConvNets) to generate the expert trajectories. As in previous work (Cazenavette et al., 2022; Zhao & Bilen, 2021; Zhao et al., 2021; Chen et al., 2023; Du et al., 2023; Cui et al., 2022), we use a 3-layer ConvNet for the lower resolution CIFAR datasets and a 4-layer ConvNet for the higher resolution TinyImageNet. Exact hyperparamters in Appendix A.

**Evaluation Setup** To test generalization in presence of limited labeled data, we evaluate encoders pre-trained on the distilled data using linear probe on 1% and 5% of downstream labeled data.

**Baselines** We compare pre-training with MKDT distilled sets to pre-training with random subsets, SAS subsets (Joshi & Mirzasoleiman, 2023), KRR-ST distilled sets (Lee et al., 2023) as well as pre-training on the full data. For KRR-ST, we use the provided distilled sets for CIFAR100 and Tiny ImageNet, and distill using the provided code for CIFAR10. We omit SAS for TinyImageNet as this subset was not provided. For distilled sets of size 5%, we consider only Random and MKDT, since other baselines did not provide distilled sets.

**Downstream Generalization Performance** Table 1 demonstrates that pre-training on CIFAR10 using MKDT with a 2% distilled set improves performance by 8% on CIFAR10 and 5% on downstream tasks over the KRR-ST baseline. Gains on CIFAR10 are consistent across 1% and 5% labeled data, but improvements on downstream tasks are more pronounced with 5%, indicating MKDT scales well with more labeled data. On CIFAR100, MKDT 2% distilled set improves performance by 6% and 8% on CIFAR100 and downstream tasks, respectively. Additionally, MKDT shows up to 3% improvement on downstream tasks for larger, higher-resolution datasets like TinyImageNet (200K examples, 64x64 resolution), highlighting MKDT's scalability. KRR-ST consistently fails to outperform SSL pre-training on random subsets across all settings. In Appendix B, we verify that this holds for fine-tuning experiments from KRR-ST (Lee et al., 2023), affirming MKDT as the only effective DD method for SSL pre-training. Table 4 shows that pre-training with larger distilled sets (5% of full data) further enhances performance by up to 13%, confirming MKDT scales effectively with distilled set size as well. Table 7 shows that MKDT outperforms the strongest baseline (random subsets) by 5% on pre-training and 7% on downstream tasks when using 10% and 50% labeled data.

Table 6: Ablation over SSL Algorithm (SimCLR), Distilled Set Size 2%

| Pre-Training Dataset | Size of Downstream Labeled Data (%) | Method | Downstream Task Accuracy | | | | | | |
|---|---|---|---|---|---|---|---|---|---|
| | | | CIFAR10 | CIFAR100 | Tiny ImageNet | Aircraft | CUB2011 | Dogs | Flowers |
| CIFAR10 | 1% | Random Subset | $35.20_{\pm1.12}$ | $7.35_{\pm0.28}$ | $2.29_{\pm0.14}$ | $2.21_{\pm0.09}$ | $1.19_{\pm0.06}$ | $1.83_{\pm0.16}$ | $1.83_{\pm0.24}$ |
| | | KRR-ST | $36.90_{\pm1.30}$ | $8.38_{\pm0.17}$ | $2.95_{\pm0.12}$ | $2.45_{\pm0.13}$ | $1.19_{\pm0.09}$ | $1.87_{\pm0.18}$ | $\mathbf{2.35}_{\pm0.06}$ |
| | | **MKDT** | $\mathbf{40.77}_{\pm1.05}$ | $\mathbf{9.17}_{\pm0.13}$ | $\mathbf{3.06}_{\pm0.16}$ | $\mathbf{2.69}_{\pm0.21}$ | $\mathbf{1.35}_{\pm0.06}$ | $\mathbf{2.02}_{\pm0.23}$ | $1.88_{\pm0.22}$ |
| | 5% | Random Subset | $45.69_{\pm0.43}$ | $15.09_{\pm0.39}$ | $5.71_{\pm0.15}$ | $5.21_{\pm1.04}$ | $1.52_{\pm0.18}$ | $2.48_{\pm0.16}$ | $3.36_{\pm0.20}$ |
| | | KRR-ST | $46.87_{\pm0.52}$ | $16.29_{\pm0.37}$ | $6.31_{\pm0.43}$ | $5.31_{\pm0.63}$ | $\mathbf{1.89}_{\pm0.14}$ | $2.66_{\pm0.18}$ | $\mathbf{4.36}_{\pm0.16}$ |
| | | **MKDT** | $\mathbf{51.77}_{\pm0.25}$ | $\mathbf{18.07}_{\pm0.52}$ | $\mathbf{6.55}_{\pm0.23}$ | $\mathbf{5.90}_{\pm0.76}$ | $\mathbf{1.89}_{\pm0.15}$ | $\mathbf{2.98}_{\pm0.36}$ | $4.09_{\pm0.28}$ |
| CIFAR100 | 1% | Random Subset | $34.67_{\pm0.89}$ | $7.35_{\pm0.54}$ | $2.29_{\pm0.16}$ | $2.23_{\pm0.21}$ | $1.10_{\pm0.06}$ | $1.78_{\pm0.26}$ | $1.77_{\pm0.13}$ |
| | | KRR-ST | $36.57_{\pm1.02}$ | $8.38_{\pm0.36}$ | $3.01_{\pm0.22}$ | $2.41_{\pm0.15}$ | $1.28_{\pm0.02}$ | $1.71_{\pm0.30}$ | $1.98_{\pm0.04}$ |
| | | **MKDT** | $\mathbf{39.59}_{\pm1.19}$ | $\mathbf{9.44}_{\pm0.37}$ | $\mathbf{3.07}_{\pm0.08}$ | $\mathbf{2.60}_{\pm0.23}$ | $\mathbf{1.33}_{\pm0.11}$ | $\mathbf{1.93}_{\pm0.27}$ | $\mathbf{2.49}_{\pm0.06}$ |
| | 5% | Random Subset | $45.67_{\pm0.69}$ | $15.11_{\pm0.44}$ | $5.21_{\pm0.29}$ | $5.21_{\pm0.67}$ | $1.51_{\pm0.14}$ | $2.54_{\pm0.19}$ | $3.37_{\pm0.47}$ |
| | | KRR-ST | $46.76_{\pm0.50}$ | $15.75_{\pm0.46}$ | $6.17_{\pm0.20}$ | $5.43_{\pm0.65}$ | $\mathbf{1.93}_{\pm0.06}$ | $2.61_{\pm0.25}$ | $3.55_{\pm0.29}$ |
| | | **MKDT** | $\mathbf{49.87}_{\pm0.75}$ | $\mathbf{18.47}_{\pm0.21}$ | $\mathbf{6.65}_{\pm0.21}$ | $\mathbf{5.56}_{\pm0.86}$ | $\mathbf{1.93}_{\pm0.13}$ | $\mathbf{2.98}_{\pm0.32}$ | $\mathbf{4.83}_{\pm0.27}$ |

Table 7: Larger Labeled Data Fractions (10%, 50%), Distilled Set Size 2%

| Pre-Training Dataset | Size of Downstream Labeled Data (%) | Method | Downstream Task Accuracy | | | | | | |
|---|---|---|---|---|---|---|---|---|---|
| | | | CIFAR10 | CIFAR100 | Tiny ImageNet | Aircraft | CUB2011 | Dogs | Flowers |
| CIFAR10 | 10% | Random Subset | $51.21_{\pm0.38}$ | $19.75_{\pm0.36}$ | $8.04_{\pm0.45}$ | $7.42_{\pm0.43}$ | $2.15_{\pm0.16}$ | $3.10_{\pm0.30}$ | $5.22_{\pm0.96}$ |
| | | KRR-ST | $51.02_{\pm0.53}$ | $21.01_{\pm0.14}$ | $8.95_{\pm0.26}$ | $7.91_{\pm0.76}$ | $2.44_{\pm0.12}$ | $3.54_{\pm0.29}$ | $6.82_{\pm0.64}$ |
| | | **MKDT** | $\mathbf{56.88}_{\pm0.85}$ | $\mathbf{24.61}_{\pm0.42}$ | $\mathbf{9.76}_{\pm0.28}$ | $\mathbf{8.96}_{\pm0.71}$ | $\mathbf{2.78}_{\pm0.22}$ | $\mathbf{4.37}_{\pm0.37}$ | $\mathbf{7.38}_{\pm1.00}$ |
| | 50% | Random Subset | $57.18_{\pm0.63}$ | $26.86_{\pm0.36}$ | $15.16_{\pm0.03}$ | $16.11_{\pm0.90}$ | $4.18_{\pm0.05}$ | $6.17_{\pm0.08}$ | $12.98_{\pm0.35}$ |
| | | KRR-ST | $58.09_{\pm0.07}$ | $29.01_{\pm0.30}$ | $15.94_{\pm0.31}$ | $17.60_{\pm1.04}$ | $5.01_{\pm0.34}$ | $6.81_{\pm0.15}$ | $15.92_{\pm0.64}$ |
| | | **MKDT** | $\mathbf{63.63}_{\pm0.17}$ | $\mathbf{34.06}_{\pm0.39}$ | $\mathbf{17.57}_{\pm0.37}$ | $\mathbf{19.43}_{\pm0.93}$ | $\mathbf{5.43}_{\pm0.22}$ | $\mathbf{7.63}_{\pm0.17}$ | $\mathbf{16.89}_{\pm0.37}$ |
| CIFAR100 | 10% | Random Subset | $52.23_{\pm0.38}$ | $21.56_{\pm0.62}$ | $8.57_{\pm0.43}$ | $8.01_{\pm0.61}$ | $2.73_{\pm0.17}$ | $3.94_{\pm0.28}$ | $7.50_{\pm1.33}$ |
| | | KRR-ST | $52.40_{\pm0.73}$ | $21.39_{\pm0.16}$ | $8.21_{\pm0.07}$ | $7.64_{\pm0.36}$ | $2.34_{\pm0.12}$ | $3.76_{\pm0.25}$ | $6.52_{\pm1.28}$ |
| | | **MKDT** | $\mathbf{57.67}_{\pm0.33}$ | $\mathbf{27.28}_{\pm0.18}$ | $\mathbf{11.05}_{\pm0.50}$ | $\mathbf{9.25}_{\pm0.70}$ | $\mathbf{3.42}_{\pm0.16}$ | $\mathbf{4.43}_{\pm0.38}$ | $\mathbf{9.35}_{\pm1.28}$ |
| | 50% | Random Subset | $60.39_{\pm0.17}$ | $31.62_{\pm0.32}$ | $16.02_{\pm0.14}$ | $18.10_{\pm0.12}$ | $5.65_{\pm0.18}$ | $7.37_{\pm0.23}$ | $16.18_{\pm0.31}$ |
| | | KRR-ST | $58.57_{\pm0.78}$ | $29.46_{\pm0.81}$ | $15.70_{\pm0.07}$ | $15.89_{\pm0.54}$ | $4.82_{\pm0.41}$ | $7.00_{\pm0.18}$ | $15.07_{\pm0.46}$ |
| | | **MKDT** | $\mathbf{65.85}_{\pm0.33}$ | $\mathbf{38.09}_{\pm0.35}$ | $\mathbf{17.46}_{\pm0.24}$ | $\mathbf{20.73}_{\pm0.24}$ | $\mathbf{6.67}_{\pm0.05}$ | $\mathbf{8.48}_{\pm0.21}$ | $\mathbf{21.15}_{\pm0.50}$ |
| TinyImageNet | 10% | Random Subset | $44.08_{\pm1.91}$ | $16.43_{\pm0.12}$ | $8.84_{\pm0.85}$ | $5.57_{\pm0.61}$ | $2.12_{\pm0.30}$ | $3.07_{\pm0.17}$ | $6.51_{\pm1.27}$ |
| | | KRR-ST | $45.48_{\pm0.84}$ | $17.02_{\pm0.26}$ | $8.88_{\pm0.41}$ | $5.29_{\pm0.16}$ | $2.08_{\pm0.21}$ | $3.21_{\pm0.15}$ | $6.10_{\pm1.44}$ |
| | | **MKDT** | $\mathbf{49.33}_{\pm0.49}$ | $\mathbf{20.19}_{\pm0.45}$ | $\mathbf{10.89}_{\pm0.63}$ | $\mathbf{6.61}_{\pm0.24}$ | $\mathbf{2.47}_{\pm0.18}$ | $\mathbf{3.86}_{\pm0.36}$ | $\mathbf{7.33}_{\pm1.34}$ |
| | 50% | Random Subset | $47.35_{\pm0.92}$ | $19.28_{\pm0.61}$ | $14.01_{\pm0.38}$ | $8.74_{\pm0.78}$ | $3.61_{\pm0.31}$ | $5.18_{\pm0.29}$ | $13.60_{\pm1.13}$ |
| | | KRR-ST | $48.16_{\pm1.18}$ | $20.01_{\pm0.40}$ | $13.59_{\pm0.47}$ | $8.66_{\pm1.29}$ | $3.46_{\pm0.22}$ | $4.98_{\pm0.31}$ | $15.12_{\pm0.37}$ |
| | | **MKDT** | $\mathbf{51.98}_{\pm0.44}$ | $\mathbf{24.41}_{\pm0.39}$ | $\mathbf{16.99}_{\pm0.25}$ | $\mathbf{10.84}_{\pm0.25}$ | $\mathbf{4.44}_{\pm0.28}$ | $\mathbf{6.22}_{\pm0.15}$ | $\mathbf{16.47}_{\pm0.46}$ |

**Ablations** We perform ablations over two factors: 1) initialization and 2) SSL algorithm. Table 2 presents results for pre-training with MKDT using random subset initialization, as well as results for pre-training directly on the high-loss subset initialization used by MKDT. Interestingly, the KD objective for SSL pre-training benefits slightly more from the high-loss subset than from random subsets. Consequently, MKDT initialized from the high-loss subset performs better than when initialized from a random subset. Table 6 shows results for MKDT using a teacher model trained with SimCLR (Chen et al., 2020) instead of BarlowTwins. Specifically, we train a ResNet-18 for 400 epochs using SimCLR. Here too, MKDT achieves approximately 6% higher performance compared to random subsets across downstream datasets. This confirms that MKDT generalizes across different SSL training algorithms.

**Generalization to Larger Architectures** Table 5 compares CIFAR100 5% size MKDT distilled set to 5% size random subsets, using the larger ResNet-10 and ResNet-18 architectures. Across all downstream tasks, we confirm that MKDT distilled sets outperform baselines even when using larger architectures. Surprisingly, the larger ResNet-18 slightly under-performs the smaller ResNet-10. This trend is observed across all baselines, including no pre-training. We conjecture this is due to larger models requiring a lot more data to be able to use their extra capacity to surpass their smaller counterparts.

## 6 CONCLUSION

To conclude, we propose MKDT, the first effective approach for dataset distillation for SSL pre-training. We demonstrated the challenges of naïvely adapting previous supervised distillation method and showed how knowledge distillation with trajectory matching can remedy these problems. Empirically, we showed up to 13% improvement in downstream accuracy when pre-training with MKDT distilled sets over the next best baseline. Thus, we enable highly data-efficient SSL pre-training.

**Acknowledgments** This research was partially supported by the National Science Foundation CA-REER Award 2146492, and an Amazon PhD fellowship.

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

# A  Experiment Details

## A.1  Additional Details for Experiments in Tables 1, 2, 3, 4, 5, 6, 7

**Training the Teacher Model Using SSL** We trained the teacher model using BarlowTwins (Zbontar et al., 2021) using the training setup ResNet18 specified in (Gedara Chaminda Bandara et al., 2023). We used the Adam optimizer with batch size 256, learning rate 0.01, cosine annealing learning rate schedule, and weight decay $10^{-6}$. The feature dimension is 1024. Finally, we use the pre-projection head representation of the trained model for teacher representation, and its dimension is of size 512.

**Training *Expert* Trajectories Using KD** We trained 100 expert trajectories for each dataset with random initialization of the network for 20 epochs, using Stochastic Gradient Descent with learning rate 0.1, momentum 0.9, and weight decay 1e-4. Similar to other DD works (Cazenavette et al., 2022; Lee et al., 2023), we used depth 4 ConvNet for Tiny ImageNet and depth 3 ConvNet for both CIFAR 10 and CIFAR 100. We did not apply any augmentation except normalization, and did not apply the ZCA-whitening.

**Distillation Hyperparameters** We distilled 2% of CIFAR 10, CIFAR 100, and Tiny ImageNet. We used SGD for optimizing the synthetic images with batch size 256, momentum 0.5. We distilled CIFAR 10 and CIFAR 100 with depth 3 ConvNet and Tiny ImageNet with depth 4 ConvNet. We initialize the synthetic learning rate as 0.1 and used SGD with learning rate $10^{-4}$ and momentum 0.5 to update it. We distilled the datasets for 5000 iterations and evaluated their performance for all the experiments except those in Table 5, where we use the distilled dataset after 1000 iterations. The other hyper-parameters are recorded in Table 8.

Table 8: MKDT Hyperparameters on 2% Distilled Set

|  | CIFAR10 | CIFAR100 | TinyImageNet |
|---|---|---|---|
| Percentage Distilled | 2% | 2% | 2% |
| Model | ConvNetD3 | ConvNetD3 | ConvNetD4 |
| Synthetic Steps ($N$) | 40 | 40 | 10 |
| Expert Epochs ($M$) | 2 | 2 | 2 |
| Max Start Epoch ($T^+$) | 5 | 2 | 2 |
| Learning Rate (Pixels) | $10^3$ | $10^3$ | $10^5$ |

**Pre-training on Synthetic Data** For the synthetic data, we pre-train them using the MSE loss with their learned representation for 20 epochs using SGD with batch size 256, their distilled learning rate, momentum 0.9, and weight decay $10^{-4}$. We use a depth 3 ConvNet for CIFAR 10 and CIFAR 100, and a depth 4 ConvNet for Tiny ImageNet. For distilled datasets, we use the synthetic learning rate $\alpha_{\text{syn}}$. For other datasets (e.g., random subset), we use the same setting except a learning rate of 0.1.

**Evaluation** We use the models' penultimate layer's representations (as is standard in contrastive learning (Zbontar et al., 2021; Chen et al., 2020)) of the downstream task's training set and train a linear classifier, using LBFGS with 100 iterations and regularization weight $10^{-3}$. We then use the pre-projection head representations of the test set of the downstream task and evaluate using the aforementioned linear classifer to report the downstream test accuracy.

**Using SimCLR for Obtaining Teacher Representation.** We conducted an ablation study using a teacher model trained with SimCLR (Chen et al., 2020) instead of Barlow Twins (Zbontar et al., 2021) for CIFAR 10 and CIFAR 100. The experiment steps are similar to the ones in A.1. During the "Training the Teacher Model Using SSL" step , we used the Adam optimizer with batch size 512, learning rate 0.001, and weight decay $10^{-6}$ to train a ResNet18 along with a 2-layer linear projection head for 400 epochs. The projection head included Batch Normalization and ReLU after the first layer, and Batch Normalization after the second layer, projecting to 128 dimensions. Then, we used the pre-projection head representation of the trained model for getting the teacher representation of size 512. The other steps are the same as the one in A.1. Table 6 shows that MKDT consistently outperforms the random subset across all downstream datasets and various sizes of labeled data, highlighting the method's generalizability to other contrastive learning methods.

**Scaling the Method to Larger Distillation Sets.** In addition to distilling 2% subsets, we also conducted experiments distilling 5% subsets of CIFAR10, CIFAR100, and TinyImageNet to evaluate the generalizability of the method to larger distillation sets. Table 4 shows the scalability of the method to larger distilled set sizes. The experiments procedure are the same as the ones in A.1 except

Table 9: MKDT Hyperparameters on 5% Distilled Set

| | CIFAR10 | CIFAR100 | TinyImageNet |
|---|---|---|---|
| Percentage Distilled | 5% | 5% | 5% |
| Model | ConvNetD3 | ConvNetD3 | ConvNetD4 |
| Synthetic Steps ($N$) | 40 | 40 | 10 |
| Expert Epochs ($M$) | 2 | 2 | 2 |
| Max Start Epoch ($T^+$) | 5 | 5 | 2 |
| Learning Rate (Pixels) | $10^4$ | $10^4$ | $10^5$ |

that we use different distillation hyperparameters for the 5% distilled set. The hyperparameters are summarized in Table 9.

**Scaling the Method to Larger Downstream Labeled Dataset Sizes.** We evaluated the performance for CIFAR 10, CIFAR 100, and TinyImageNet on larger downstream labeled data sizes, specifically 10% and 50% labeled data sizes, using the 2% distilled set obtained with the method illustrated in A.1. As shown in Table 7, MKDT continues to outperform random subset across all downstream datasets and data sizes, demonstrating its scalability with larger labeled data sizes.

**Details for KRR-ST Lee et al. (2023)** We use the code and hyper-parameters provided in (Lee et al., 2023). As the original paper did not provide the results and the hyperparameters for CIFAR 10, we use the same hyperparameters as CIFAR 100 to distill CIFAR 10. In particular, this invovles using BarlowTwins ResNet-18 as the teacher model as well.

## A.2    ADDITIONAL DETAILS ON EXPERIMENTS IN FIG. 1

For the experiment in Figure 1a, we train 5 trajectories of each of MTT SSL and MTT SL for CIFAR 100 using the same random initialization of the network, respectively. For MTT SSL, we train the models with the Adam optimizer with batch size 1024, learning rate 0.001, and weight decay $10^{-6}$. For MTT SL, we train the model with SGD with batch size 256, learning rate 0.01, momentum 0, and no weight decay.

For both of the experiments in Figure 1b and 1c, we distill the dataset using MTT SL with image learning rate 1000, max start epoch 0, synthetic steps 20, and expert epochs 4. We distill using MTT SLL with image learning rate 1000, max start epoch 0, synthetic steps 10, and expert epochs 2. We distilled them for 100 iterations and record the change in the loss function and the average absolute change in pixels.

# B  ADDITIONAL COMPARISON OF RANDOM SUBSET SSL PRE-TRAINING TO KRR-ST

This experiment is conducted in the setting of Lee et al. (2023) for pre-training on CIFAR100. This experiment pre-trains on a distilled set / subset of 2% the size of CIFAR100 and then evaluates on downstream tasks by finetuning the entire network with the entire labeled dataset for the downstream task. KRR-ST compares with the random subset baseline by pre-training using SL. However, we use SSL on the random subset as a baseline instead. Here, we show that, in fact, even in the finetuning setting reported by Lee et al. (2023), KRR-ST cannot outperform SSL pre-training on random real images. The baseline reported as 'Random' in Lee et al. (2023) refers to SL pre-training as opposed to SSL pre-training.

Table 10: Pre-training on CIFAR100

| Method | Pre-Training | Downstream | | | | | |
|---|---|---|---|---|---|---|---|
| | CIFAR100 | CIFAR10 | Aircraft | Cars | CUB2011 | Dogs | Flowers |
| No pre-training (from Lee et al. (2023)) | $64.95_{\pm0.21}$ | $87.34_{\pm0.13}$ | $34.66_{\pm0.39}$ | $19.43_{\pm0.14}$ | $18.46_{\pm0.11}$ | $22.31_{\pm0.22}$ | $58.75_{\pm0.41}$ |
| Random (Supervised Learning (from Lee et al. (2023)) | $65.23_{\pm0.12}$ | $87.55_{\pm0.19}$ | $33.99_{\pm0.45}$ | $19.77_{\pm0.21}$ | $18.18_{\pm0.21}$ | $21.69_{\pm0.18}$ | $59.31_{\pm0.27}$ |
| KRR-ST (from Lee et al. (2023)) | $66.81_{\pm0.11}$ | $88.72_{\pm0.11}$ | $41.54_{\pm0.37}$ | $28.68_{\pm0.32}$ | $25.30_{\pm0.37}$ | $26.39_{\pm0.08}$ | $67.88_{\pm0.18}$ |
| **Random (SSL Pre-training)** | $66.44_{\pm0.14}$ | $88.74_{\pm0.20}$ | $42.02_{\pm0.06}$ | $28.75_{\pm0.23}$ | $25.12_{\pm0.19}$ | $26.57_{\pm0.22}$ | $68.21_{\pm0.40}$ |

## C   PROOF FOR THEOREM 4.1

*Proof.* To analyze the variance of the gradient for both SL and SSL, we will conduct a case analysis. There are 3 possible unique cases for constructing the mini-batch. Assuming $n$ is extremely large s.t. $\frac{n/2}{n} \approx \frac{n/2-1}{n}$, we have:

1. Both examples are from class 0 with probability = $\frac{1}{4}$

2. Both examples are from class 1 with probability = $\frac{1}{4}$

3. 1 example from each class with probability = $\frac{1}{2}$

Let $x_1, y_1$ refer to example 1 and its corresponding label vector; similarly, let $x_2, y_2$ refer to example 2 and its corresponding label vector.

To show the effect of each term, we refer to the class mean vectors for class 0 and class 1 as $\mu_0$ and $\mu_1$, respectively; and use $e_0$ and $e_1$, the basis vectors to represent the labels for class 0 and class 1.

**Analyzing cases for SL**

$$\nabla_W(L_{SL}(B)) = \frac{1}{2} \sum_{i=1}^{2} 2(Wx_i - y_i)x_i^\top \tag{10}$$

$$= (Wx_1 - y_1)(x_1)^\top + (Wx_2 - y_2)(x_2)^\top \tag{11}$$

*Case 1: Both examples from class 0*

$$\nabla_W(L_{SL}(B)) = (Wx_1 - e_0)(x_1)^\top + (Wx_2 - e_0)(x_2)^\top \tag{12}$$

$$\mathbb{E}[\nabla_W(L_{SL}(B))] = \mathbb{E}[(Wx_1 - e_0)(x_1)^\top + (Wx_2 - e_0)(x_2)^\top] \tag{13}$$

$$= \mathbb{E}[(Wx_1 - e_0)(x_1)^\top] + \mathbb{E}[Wx_2 - e_0)(x_2)^\top] \tag{14}$$

$$= \mathbb{E}[(W\mu_0 + W\epsilon_{N_1} - e_0)(e_0 + \epsilon_{N_1})^\top] + \mathbb{E}[(W\mu_0 + W\epsilon_{N_2} - e_0)(e_0 + \epsilon_{N_2})^\top] \tag{15}$$

$$= \mathbb{E}[(\mu_0 + \epsilon_{N_1} - e_0)(\mu_0 + \epsilon_{N_1})^\top] + \mathbb{E}[(\mu_0 + \epsilon_{N_2} - e_0)(\mu_0 + \epsilon_{N_2})^\top] \text{by substituting } W = I \tag{16}$$

$$= \mathbb{E}[\mu_0\mu_0^\top + \epsilon_{N_1}\mu_0^\top - e_0\mu_0^\top + \mu_0\epsilon_{N_1}^\top + \epsilon_{N_1}\epsilon_{N_1}^\top - e_0\epsilon_{N_1}^\top]$$
$$+ \mathbb{E}[\mu_0\mu_0^\top + \epsilon_{N_2}\mu_0^\top - e_0\mu_0^\top + \mu_0\epsilon_{N_2}^\top + \epsilon_{N_2}\epsilon_{N_2}^\top - e_0\epsilon_{N_2}^\top] \tag{17}$$

$$= 2(\mu_0\mu_0^\top - e_0\mu_0^T) \tag{18}$$

$$\nabla_W(L_{SL}(B)) = \mu_0\mu_0^\top + \epsilon_{N_1}\mu_0^\top - e_0\mu_0^\top + \mu_0\epsilon_{N_1}^\top + \epsilon_{N_1}\epsilon_{N_1}^\top - e_0\epsilon_{N_1}^\top$$
$$+ \mu_0\mu_0^\top + \epsilon_{N_2}\mu_0^\top - e_0\mu_0^\top + \mu_0\epsilon_{N_2}^\top + \epsilon_{N_2}\epsilon_{N_2}^\top - e_0\epsilon_{N_2}^\top$$
$$= 2\mu_0\mu_0^\top + (\epsilon_{N_1} + \epsilon_{N_2})\mu_0^\top - 2e_0\mu_0^\top + \mu_0(\epsilon_{N_1}^\top + \epsilon_{N_2}^\top)$$
$$+ \epsilon_{N_1}\epsilon_{N_1}^\top + \epsilon_{N_2}\epsilon_{N_2}^\top - e_0(\epsilon_{N_1}^\top + \epsilon_{N_2}^\top) \tag{19}$$

*Case 2: Both examples from class 1*

By symmetry, we have:

$$\mathbb{E}[\nabla_W(L_{SL}(B))] = 2(\mu_1\mu_1^\top - e_1\mu_1^\top) \tag{20}$$

and

$$\nabla_W L_{SL}(B) = 2\mu_1\mu_1^\top + (\epsilon_{N_1} + \epsilon_{N_2})\mu_1^\top - 2e_1\mu_1^\top + \mu_1(\epsilon_{N_1}^\top + \epsilon_{N_2}^\top)$$
$$+ \epsilon_{N_1}\epsilon_{N_1}^\top + \epsilon_{N_2}\epsilon_{N_2}^\top - e_1(\epsilon_{N_1}^\top + \epsilon_{N_2}^\top) \tag{21}$$

*Case 3: 1 example from each class*

$$\mathbb{E}[\nabla_W(L_{SL}(B))] = \mu_0\mu_0^\top - e_0\mu_0^\top + \mu_1\mu_1^\top - e_1\mu_1^\top \tag{22}$$

and

$$\nabla_W L_{SL}(B) = \mu_0\mu_0^\top + \epsilon_{N_1}\mu_0^\top - e_0\mu_0^\top + \mu_0\epsilon_{N_1}^\top + \epsilon_{N_1}\epsilon_{N_1}^\top - e_0\epsilon_{N_1}^\top$$
$$+ \mu_1\mu_1^\top + \epsilon_{N_2}\mu_1^\top - e_1\mu_1^\top + \mu_1\epsilon_{N_2}^\top + \epsilon_{N_2}\epsilon_{N_2}^\top - e_1\epsilon_{N_2}^\top \tag{23}$$

*Putting it together*

$$\mathbb{E}[\nabla_W(L_{SL}(B))] = \frac{1}{4} \cdot 2(\mu_0\mu_0^\top - e_0\mu_0^\top) \tag{24}$$
$$+ \frac{1}{4} \cdot 2(\mu_1\mu_1^\top - e_1\mu_1^\top) \tag{25}$$
$$+ \frac{1}{2}\left(\mu_0\mu_0^\top - e_0\mu_0^\top + \mu_1\mu_1^\top - e_1\mu_1^\top\right) \tag{26}$$
$$= \mu_0\mu_0^\top - e_0\mu_0^\top + \mu_1\mu_1^\top - e_1\mu_1^\top \tag{27}$$

Finally,

$$Var(\nabla_W(L_{SL}(B))) = \mathbb{E}[\|\nabla_W(L_{SL}(B)) - \mathbb{E}[\nabla_W(L_{SL}(B))]\|^2] \tag{28}$$
$$= \frac{1}{4}\mathbb{E}[(\|\nabla_W(L_{SL}(B)) - \mathbb{E}[\nabla_W(L_{SL}(B))]\|^2)|\text{case 1}]$$
$$+ \frac{1}{4}\mathbb{E}[(\|\nabla_W(L_{SL}(B)) - \mathbb{E}[\nabla_W(L_{SL}(B))]\|^2)|\text{case 2}]$$
$$+ \frac{1}{2}\mathbb{E}[(\|\nabla_W(L_{SL}(B)) - \mathbb{E}[\nabla_W(L_{SL}(B))]\|^2)|\text{case 3}] \tag{29}$$

*Simplifying term 1*: $\mathbb{E}[(\|\nabla_W(L_{SL}(B)) - \mathbb{E}[\nabla_W(L_{SL}(B))]\|^2)|\text{case 1}]$

$$\nabla_W(L_{SL}(B)) - \mathbb{E}[\nabla_W(L_{SL}(B))] = 2\mu_0\mu_0^\top + (\epsilon_{N_1} + \epsilon_{N_2})\mu_0^\top - 2e_0\mu_0^\top + \mu_0(\epsilon_{N_1}^\top + \epsilon_{N_2}^\top)$$
$$+ \epsilon_{N_1}\epsilon_{N_1}^\top + \epsilon_{N_2}\epsilon_{N_2}^\top - e_0(\epsilon_{N_1}^\top + \epsilon_{N_2}^\top)$$
$$- \left(\mu_0\mu_0^\top - e_0\mu_0^\top + \mu_1\mu_1^\top - e_1\mu_1^\top\right)$$
$$= \mu_0\mu_0^\top + \epsilon_{N_1}\mu_0^\top + \epsilon_{N_2}\mu_0^\top - e_0\mu_0^\top + \mu_0\epsilon_{N_1}^\top + \mu_0\epsilon_{N_2}^\top$$
$$+ \epsilon_{N_1}\epsilon_{N_1}^\top + \epsilon_{N_2}\epsilon_{N_2}^\top - e_0\epsilon_{N_1}^\top - e_0\epsilon_{N_2}^\top$$
$$- \mu_1\mu_1^\top + e_1\mu_1^\top \tag{30}$$

*By symmetry, term 2* i.e. $\mathbb{E}[(\|\nabla_W(L_{SL}(B)) - \mathbb{E}[\nabla_W(L_{SL}(B))]\|^2)|\text{case 2}]$ can be simplified as follows:

$$\nabla_W(L_{SL}(B)) - \mathbb{E}[\nabla_W(L_{SL}(B))] = \mu_1\mu_1^\top + \epsilon_{N_1}\mu_1^\top + \epsilon_{N_2}\mu_1^\top - e_1\mu_1^\top + \mu_1\epsilon_{N_1}^\top + \mu_1\epsilon_{N_2}^\top$$

$$+ \epsilon_{N_1} \epsilon_{N_1}^\top + \epsilon_{N_2} \epsilon_{N_2}^\top - e_1 \epsilon_{N_1}^\top - e_1 \epsilon_{N_2}^\top$$
$$- \mu_0 \mu_0^\top + e_0 \mu_0^\top \tag{31}$$

*Simplifying term 3*:
$$\mathbb{E}[(\|\nabla_W(L_{SL}(B)) - \mathbb{E}[\nabla_W(L_{SL}(B))]\|^2)|\text{case 3}] \tag{32}$$

$$\begin{aligned}
\nabla_W(L_{SL}(B)) - \mathbb{E}[\nabla_W(L_{SL}(B))] &= \mu_0 \mu_0^\top + \epsilon_{N_1} \mu_0^\top - e_0 \mu_0^\top + \mu_0 \epsilon_{N_1}^\top + \epsilon_{N_1} \epsilon_{N_1}^\top - e_0 \epsilon_{N_1}^\top \\
&\quad + \mu_1 \mu_1^\top + \epsilon_{N_2} \mu_1^\top - e_1 \mu_1^\top + \mu_1 \epsilon_{N_2}^\top + \epsilon_{N_2} \epsilon_{N_2}^\top - e_1 \epsilon_{N_2}^\top \\
&\quad - \left( \mu_0 \mu_0^\top - e_0 \mu_0^\top + \mu_1 \mu_1^\top - e_1 \mu_1^\top \right) \\
&= \epsilon_{N_1} \mu_0^\top + \mu_0 \epsilon_{N_1}^\top + \epsilon_{N_1} \epsilon_{N_1}^\top - e_0 \epsilon_{N_1}^\top \\
&\quad + \epsilon_{N_2} \mu_1^\top + \mu_1 \epsilon_{N_2}^\top + \epsilon_{N_2} \epsilon_{N_2}^\top - e_1 \epsilon_{N_2}^\top
\end{aligned} \tag{33}$$

**Analyzing cases for SSL**

We will now analyze the same for SSL and show by comparing each of the 3 terms, element-wise, above to their counterparts that $Var(L_{SSL}(B)) > Var(L_{SL}(B))$

From Xue et al. (2023), we have that $L_{SSL}$ can be re-written as
$$\nabla_W L_{SSL}(W) = -\text{Tr}(2\tilde{M}WW^\top) + \text{Tr}(MW^\top W M W^\top W)$$

where
$$M = \frac{1}{2m} \sum_{i=1}^{2m} x_i x_i^\top$$

with $m$ being the number of augmentations, and $M$ represents the covariance of the training data. The matrix $\tilde{M}$ is defined as:
$$\tilde{M} = \frac{1}{2} \sum_{i=1}^{n} \left( \frac{1}{m} \sum_{x \in \mathcal{A}(x_i)} x \right) \left( \frac{1}{m} \sum_{x \in \mathcal{A}(x_i)} x^\top \right)$$

where $\mathcal{A}(x_i)$ denotes the set of augmentations for the sample $x_i$.

As $m \to \infty$, $M = \tilde{M} = \frac{1}{2}\left( x_1 x_1^T + x_2 x_2^T \right)$.

Hence, $\nabla_W(L_{SSL}(B)) = -4WM + 4WMW^\top WM$

Substituting $W = I$, we get $\nabla_W(L_{SSL}(B)) = -4M + 4M^2$

*Case 1: Both examples from class 0*

$$\begin{aligned}
M &= \frac{1}{2}\left( x_1 x_1^\top + x_2 x_2^\top \right) \tag{34} \\
&= \frac{1}{2}\left( (\mu_0 + \epsilon_{N_1})(\mu_0 + \epsilon_{N_1})^\top + (\mu_0 + \epsilon_{N_2})(\mu_0 + \epsilon_{N_2})^\top \right) \tag{35} \\
&= \frac{1}{2}\big( \mu_0 \mu_0^\top + \mu_0 \epsilon_{N_1}^\top + \epsilon_{N_1} \mu_0^\top + \epsilon_{N_1} \epsilon_{N_1}^\top \\
&\quad + \mu_0 \mu_0^\top + \mu_0 \epsilon_{N_2}^\top + \epsilon_{N_2} \mu_0^\top + \epsilon_{N_2} \epsilon_{N_2}^\top \big) \tag{36} \\
&= \mu_0 \mu_0^\top + \frac{1}{2}\big( \mu_0 \epsilon_{N_1}^\top + \epsilon_{N_1} \mu_0^\top + \epsilon_{N_1} \epsilon_{N_1}^\top \\
&\quad + \mu_0 \epsilon_{N_2}^\top + \epsilon_{N_2} \mu_0^\top + \epsilon_{N_2} \epsilon_{N_2}^\top \big) \tag{37}
\end{aligned}$$

$$M^2 = \left( \mu_0 \mu_0^\top + \frac{1}{2}\big( \mu_0 \epsilon_{N_1}^\top + \epsilon_{N_1} \mu_0^\top + \epsilon_{N_1} \epsilon_{N_1}^\top \right. \tag{38}$$

$$+ \mu_0 \epsilon_{N_2}^\top + \epsilon_{N_2} \mu_0^\top + \epsilon_{N_2} \epsilon_{N_2}^\top))^2 \tag{39}$$

$$
\begin{aligned}
= \ & \mu_0 \mu_0^\top \mu_0 \mu_0^\top \\
& + \mu_0 \mu_0^\top \cdot \frac{1}{2} \left( \mu_0 \epsilon_{N_1}^\top + \epsilon_{N_1} \mu_0^\top + \epsilon_{N_1} \epsilon_{N_1}^\top \right) \\
& + \mu_0 \mu_0^\top \cdot \frac{1}{2} \left( \mu_0 \epsilon_{N_2}^\top + \epsilon_{N_2} \mu_0^\top + \epsilon_{N_2} \epsilon_{N_2}^\top \right) \\
& + \frac{1}{2} \left( \mu_0 \epsilon_{N_1}^\top + \epsilon_{N_1} \mu_0^\top + \epsilon_{N_1} \epsilon_{N_1}^\top \right) \mu_0 \mu_0^\top \\
& + \frac{1}{2} \left( \mu_0 \epsilon_{N_2}^\top + \epsilon_{N_2} \mu_0^\top + \epsilon_{N_2} \epsilon_{N_2}^\top \right) \mu_0 \mu_0^\top \\
& + \frac{1}{4} \left( \mu_0 \epsilon_{N_1}^\top + \epsilon_{N_1} \mu_0^\top + \epsilon_{N_1} \epsilon_{N_1}^\top \right)^2 \\
& + \frac{1}{4} \left( \mu_0 \epsilon_{N_2}^\top + \epsilon_{N_2} \mu_0^\top + \epsilon_{N_2} \epsilon_{N_2}^\top \right)^2 \\
& + \frac{1}{4} \left( \mu_0 \epsilon_{N_1}^\top + \epsilon_{N_1} \mu_0^\top + \epsilon_{N_1} \epsilon_{N_1}^\top \right) \left( \mu_0 \epsilon_{N_2}^\top + \epsilon_{N_2} \mu_0^\top + \epsilon_{N_2} \epsilon_{N_2}^\top \right) \\
& + \frac{1}{4} \left( \mu_0 \epsilon_{N_2}^\top + \epsilon_{N_2} \mu_0^\top + \epsilon_{N_2} \epsilon_{N_2}^\top \right) \left( \mu_0 \epsilon_{N_1}^\top + \epsilon_{N_1} \mu_0^\top + \epsilon_{N_1} \epsilon_{N_1}^\top \right)
\end{aligned}
\tag{40}
$$

$$\mathbb{E}[-4M + 4MM] = -4\mu_0\mu_0^\top + 4\mu_0\mu_0^\top \mu_0\mu_0^\top \tag{41}$$

*Case 2: Both examples from class 1*

By symmetry,

$$
\begin{aligned}
M = \ & \mu_1 \mu_1^\top + \frac{1}{2} \big( \mu_1 \epsilon_{N_1}^\top + \epsilon_{N_1} \mu_1^\top + \epsilon_{N_1} \epsilon_{N_1}^\top \\
& + \mu_1 \epsilon_{N_2}^\top + \epsilon_{N_2} \mu_1^\top + \epsilon_{N_2} \epsilon_{N_2}^\top \big)
\end{aligned}
\tag{42}
$$

$$\mathbb{E}[-4M + 4MM] = -4\mu_1\mu_1^\top + 4\mu_1\mu_1^\top \mu_1\mu_1^\top \tag{43}$$

*Case 3: 1 example from each class*

$$
\begin{aligned}
M &= \frac{1}{2} \left( x_1 x_1^\top + x_2 x_2^\top \right) \tag{44} \\
&= \frac{1}{2} \left( (\mu_0 + \epsilon_{N_1})(\mu_0 + \epsilon_{N_1})^\top + (\mu_1 + \epsilon_{N_2})(\mu_1 + \epsilon_{N_2})^\top \right) \tag{45} \\
&= \frac{1}{2} \big( \mu_0 \mu_0^\top + \mu_0 \epsilon_{N_1}^\top + \epsilon_{N_1} \mu_0^\top + \epsilon_{N_1} \epsilon_{N_1}^\top \\
&\qquad + \mu_1 \mu_1^\top + \mu_1 \epsilon_{N_2}^\top + \epsilon_{N_2} \mu_1^\top + \epsilon_{N_2} \epsilon_{N_2}^\top \big) \tag{46}
\end{aligned}
$$

$$\mathbb{E}[-4M + 4MM] = -2\mu_0\mu_0^\top - 2\mu_1\mu_1^\top + 2\mu_0\mu_0^\top \mu_0\mu_0^\top + 2\mu_1\mu_1^\top \mu_1\mu_1^\top \tag{47}$$

*Putting it together*

Hence.

$$
\begin{aligned}
\mathbb{E}[-4M + 4M^2] = \ & \frac{1}{4} \left( -4\mu_0\mu_0^\top + 4\mu_0\mu_0^\top \mu_0\mu_0^\top \right) \\
& + \frac{1}{4} \left( -4\mu_1\mu_1^\top + 4\mu_1\mu_1^\top \mu_1\mu_1^\top \right)
\end{aligned}
$$

$$+ \frac{1}{2}\left(-2\mu_0\mu_0^\top - 2\mu_1\mu_1^\top + 2\mu_0\mu_0^\top\mu_0\mu_0^\top + 2\mu_1\mu_1^\top\mu_1\mu_1^\top\right) \qquad (48)$$

$$= -2\mu_0\mu_0^\top - 2\mu_1\mu_1^\top + 2\mu_0\mu_0^\top\mu_0\mu_0^\top + 2\mu_1\mu_1^\top\mu_1\mu_1^\top \qquad (49)$$

**Comparing each term, element-wise**

*Comparing Term 1*

$$\nabla_W(L_{SL}(B)) - \mathbb{E}[\nabla_W(L_{SL}(B))] = \mu_0\mu_0^\top + \epsilon_{N_1}\mu_0^\top + \epsilon_{N_2}\mu_0^\top - e_0\mu_0^\top + \mu_0\epsilon_{N_1}^\top + \mu_0\epsilon_{N_2}^\top$$
$$+ \epsilon_{N_1}\epsilon_{N_1}^\top + \epsilon_{N_2}\epsilon_{N_2}^\top - e_0\epsilon_{N_1}^\top - e_0\epsilon_{N_2}^\top$$
$$- \mu_1\mu_1^\top + e_1\mu_1^\top \qquad (50)$$

$$\nabla_W(L_{SSL}(B)) - \mathbb{E}[\nabla_W(L_{SSL}(B))] = -2\mu_0\mu_0^\top - 2\mu_0\epsilon_{N_1}^\top - 2\epsilon_{N_1}\mu_0^\top - 2\epsilon_{N_1}\epsilon_{N_1}^\top$$
$$- 2\mu_0\epsilon_{N_2}^\top - 2\epsilon_{N_2}\mu_0^\top - 2\epsilon_{N_2}\epsilon_{N_2}^\top$$
$$= 2\mu_0\mu_0^\top\mu_0\mu_0^\top$$
$$+ \mu_0\mu_0^\top \cdot 2\left(\mu_0\epsilon_{N_1}^\top + \epsilon_{N_1}\mu_0^\top + \epsilon_{N_1}\epsilon_{N_1}^\top\right)$$
$$+ \mu_0\mu_0^\top \cdot 2\left(\mu_0\epsilon_{N_2}^\top + \epsilon_{N_2}\mu_0^\top + \epsilon_{N_2}\epsilon_{N_2}^\top\right)$$
$$+ 2\left(\mu_0\epsilon_{N_1}^\top + \epsilon_{N_1}\mu_0^\top + \epsilon_{N_1}\epsilon_{N_1}^\top\right)\mu_0\mu_0^\top$$
$$+ 2\left(\mu_0\epsilon_{N_2}^\top + \epsilon_{N_2}\mu_0^\top + \epsilon_{N_2}\epsilon_{N_2}^\top\right)\mu_0\mu_0^\top$$
$$+ 1\left(\mu_0\epsilon_{N_1}^\top + \epsilon_{N_1}\mu_0^\top + \epsilon_{N_1}\epsilon_{N_1}^\top\right)^2$$
$$+ 1\left(\mu_0\epsilon_{N_2}^\top + \epsilon_{N_2}\mu_0^\top + \epsilon_{N_2}\epsilon_{N_2}^\top\right)^2$$
$$+ 1\left(\mu_0\epsilon_{N_1}^\top + \epsilon_{N_1}\mu_0^\top + \epsilon_{N_1}\epsilon_{N_1}^\top\right)\left(\mu_0\epsilon_{N_2}^\top + \epsilon_{N_2}\mu_0^\top + \epsilon_{N_2}\epsilon_{N_2}^\top\right)$$
$$+ 1\left(\mu_0\epsilon_{N_2}^\top + \epsilon_{N_2}\mu_0^\top + \epsilon_{N_2}\epsilon_{N_2}^\top\right)\left(\mu_0\epsilon_{N_1}^\top + \epsilon_{N_1}\mu_0^\top + \epsilon_{N_1}\epsilon_{N_1}^\top\right)$$
$$(51)$$

Recall that $\mu_0 = e_0$.

Use $\epsilon_1$ and $\epsilon_2$ to denote the value of an abtrirary element of $\epsilon_{N_1}$ and $\epsilon_{N_2}$, respectively.

*Comparing the expectation of the square of element $(0,0)$ of $\nabla_W(L(B)) - \mathbb{E}[\nabla_W(L(B))]$:*

For SL: $\mathbb{E}[(1 + \epsilon_1 + \epsilon_2 - 1 + \epsilon_1 + \epsilon_2 + \epsilon_1^2 + \epsilon_2^2 - \epsilon_1 - \epsilon_2 - 1 + 1)^2] = \mathbb{E}[(\epsilon_1 + \epsilon_2 + \epsilon_1^2 + \epsilon_2^2)] =$

From Eq. 51, it is easy to see that the expectation of the square of element $(0,0)$ is far larger due to far more $\epsilon_1$, $\epsilon_2$, $\epsilon_1^2$ and $\epsilon_2^2$ terms.

Similar argument holds for diagonal terms that are not $(0,0)$, due to the greater number of terms that only have $\epsilon_1^2$ and $\epsilon_2^2$ (all terms having only $\epsilon_1$ and $\epsilon_2$ are only present in element $(0,0)$ since they are always multiplied with $e_0$).

An analogous argument holds for off-diagonal elements. Here, the contribution is only due to terms containing only $\epsilon_{N_1}$ and/or $\epsilon_{N_2}$.

*Comparing Term 2*

By symmetry, it must be that the same holds for term 2.

*Comparing Term 3*

$$\nabla_W(L_{SL}(B)) - \mathbb{E}[\nabla_W(L_{SL}(B))] = \epsilon_{N_1}\mu_0^\top + \mu_0\epsilon_{N_1}^\top + \epsilon_{N_1}\epsilon_{N_1}^\top - e_0\epsilon_{N_1}^\top$$
$$+ \epsilon_{N_2}\mu_1^\top + \mu_1\epsilon_{N_2}^\top + \epsilon_{N_2}\epsilon_{N_2}^\top - e_1\epsilon_{N_2}^\top \qquad (52)$$

$$\nabla_W(L_{SSL}(B)) - \mathbb{E}[\nabla_W(L_{SSL}(B))] = -2\left(\mu_0\mu_0^\top + \mu_0\epsilon_{N_1}^\top + \epsilon_{N_1}\mu_0^\top + \epsilon_{N_1}\epsilon_{N_1}^\top\right.$$

$$+ \mu_1 \mu_1^\top + \mu_1 \epsilon_{N_2}^\top + \epsilon_{N_2} \mu_1^\top + \epsilon_{N_2} \epsilon_{N_2}^\top \big)$$
$$+ 4 \big( \mu_0 \mu_0^\top + \mu_0 \epsilon_{N_1}^\top + \epsilon_{N_1} \mu_0^\top + \epsilon_{N_1} \epsilon_{N_1}^\top$$
$$+ \mu_1 \mu_1^\top + \mu_1 \epsilon_{N_2}^\top + \epsilon_{N_2} \mu_1^\top + \epsilon_{N_2} \epsilon_{N_2}^\top \big)^2 \qquad (53)$$

Again, from analogous aruguments from *Term 1*, we can see that the expression for $L_{SSL}$ has larger expected square element-wise than that of $L_{SL}$.

**Conclusion**

Since, for each term (corresponding to the 3 cases of mini-batch), we have that

$$\mathbb{E}[\|\nabla_W(L_{SSL}(B)) - \mathbb{E}[\nabla_W(L_{SSL}(B))]\|^2] > \mathbb{E}[\|\nabla_W(L_{SL}(B)) - \mathbb{E}[\nabla_W(L_{SL}(B))]\|^2], \quad (54)$$

Thus, we can conclude

$$\mathrm{Var}(\nabla_W L_{SL}(B)) < \mathrm{Var}(\nabla_W L_{SSL}(B)).$$

$\square$

# D  PROPOSITION D.1: GENERALIZED ANALYSIS OF VARIANCE OF TRAJECTORY UNDER SYNCHRONOUS PARALLEL SGD

**Proposition D.1.** *Let $D = \{(x_i, y_i)\}_{i=1}^N$ be a dataset with $N$ examples, where $x_i$ is the $i$-th input and $y_i \in \{1, 2, \ldots, K\}$ is its corresponding class label from $K$ classes. We assume the data $x_i$ is generated from a distribution where each example from class $k$ can be modeled as a $d$-dimensional $(k \ll d)$ 1-hot vector $e_k$ plus some noise $\epsilon_i$, i.e. $x_i = e_{y_i} + \epsilon_i$, where $\mu_{y_i}$ is the vector corresponding to class $y_i$ and $\epsilon_i \sim \mathcal{N}(0, I)$. Moreover, we consider a linear model $f_\theta(x) = \theta x$ where $\theta \in \mathbb{R}^{K \times d}$.*

*The supervised loss function $L_{SL}$ is defined as a loss function that depends solely on each individual example:*

$$L_{SL}(B) = \frac{1}{|B|} \sum_{i \in B} \ell_{SL}(f_\theta(x_i), y_i), \tag{55}$$

*where $\ell_{SL}$ is an arbitrary supervised loss function (e.g., MSE, Cross-Entropy, etc.) and $B$ is a mini-batch of examples.*

*For SSL, we consider the spectral contrastive loss:*

$$L_{SSL} = -2\mathbb{E}_{\substack{x_1,x_2 \sim \mathcal{A}(x_i) \\ x_i \sim B}} \left[ f(x_1)^T f(x_2) \right] + \mathbb{E}_{x_i, x_j \sim \mathcal{A}(B)} \left[ f(x_i)^T f(x_j) \right]^2 \tag{56}$$

*where $\mathcal{A}$ denotes the augmentation that are modeled as: $\mathcal{A}(x_i) \sim x_i + \epsilon_{aug}$, where $\epsilon_{aug} \sim \mathcal{N}(0, I)$.*

*We analyze this setting under synchronous parallel SGD, where the number of mini-batches equals the number of parallel threads. Let $P$ represent a partition of the dataset $D$ into a set of disjoint mini-batches, each of size $|B|$. The synchronous parallel SGD update for an arbitrary loss function $L(B)$ on mini-batch $B$ at time step $t$ is $\theta_{t+1} := \theta_t - \alpha \sum_{B \in P} \nabla_{\theta_t} L(B)$.*

*Let $\theta$ be the initial parameters and $\theta_{SL}$, $\theta_{SSL}$ be the model parameters after a single epoch of training with synchronous parallel using the supervised loss $L_{SL}$ and the self-supervised loss $L_{SSL}$, respectively, with a learning rate $\alpha$. Then, for sufficiently large batch size $|B|$,*

$$Var_P(\theta_{SL}) < Var_P(\theta_{SSL}), \tag{57}$$

*where $Var_P$ represents the variance over different partitions $P$, and for a vector random variable $X$, we define the scalar variance as $Var(X) = \mathbb{E}[\|X - \mathbb{E}[X]\|^2]$*

*Proof.* For synchronous parallel SGD, we have:

$$\theta_{t+1} = \theta_t - \alpha \sum_{B \in P} \nabla_{\theta_t} L(B),$$

where $L$ is an arbitrary loss function.

**Variance of Trajectory in Supervised Learning**

Consider two arbitrary partitions $P_1$ and $P_2$. Let $\theta$ represent the initial parameters, and let $\theta_{SL_1}$ and $\theta_{SL_2}$ represent the parameters after a single epoch of synchronous parallel SGD for supervised learning using $L_{SL}$ on the partitions $P_1$ and $P_2$, respectively.

We can express the updates as:

$$\theta_{SL_1} = \theta + \alpha \sum_{B \in P_1} \nabla_\theta L_{SL}(B) \tag{58}$$

$$= \theta + \alpha \sum_{B \in P_1} \sum_{(x_i, y_i) \in B} \frac{1}{|B|} \nabla_\theta L_{SL}(\{x_i, y_i\}) \text{ (due to independence of loss for each example)} \tag{59}$$

$$= \theta + \alpha \sum_{(x_i, y_i) \in D} \frac{1}{|B|} \nabla_\theta L_{SL}(\{x_i, y_i\}) \tag{60}$$

$$= \theta + \alpha \sum_{B \in P_2} \sum_{(x_i, y_i) \in B} \frac{1}{|B|} \nabla_\theta L_{SL}(\{x_i, y_i\}) \tag{61}$$

$$= \theta_{SL_2}. \tag{62}$$

Since $\theta_{SL_1} = \theta_{SL_2}$ for arbitrary partitions $P_1$ and $P_2$, we can conclude that:

$$\text{Var}_P(\theta_{SL}) = 0.$$

**Variance of Trajectory in Self-Supervised Learning**

Next, we need to show that $\text{Var}_P(\theta_{SSL}) > 0$. Let $\theta$ represent the initial parameters, and let $\theta_{SSL_1}$ and $\theta_{SSL_2}$ represent the parameters after a single epoch of synchronous parallel SGD using the self-supervised loss $L_{SSL}$ with partitions $P_1$ and $P_2$, respectively.

To show $\text{Var}_P(\theta_{SSL}) > 0$, it suffices to show that there exist partitions $P_1$ and $P_2$ such that $\theta_{SSL_1} \neq \theta_{SSL_2}$.

Consider the following two partitions:

- Let $P_1$ be a partition where each mini-batch is constructed by iterating over classes, until $|B|$ examples are selected, and picking a single example from the remaining examples in each class

- Let $P_2$ be a partition where each mini-batch is constructed by picking examples from the remaining examples each class until $|B|$ examples

It is easy to see, from prior work in hard negative mining for contrastive SSL such as Robinson et al. (2021), that $P_2$ will have a different gradient across all batches from $P_1$ as $P_2$ only contrasts examples within the same class, whereas $P_1$ contrasts examples across different classes. Hence, $\theta_{SSL_1} \neq \theta_{SSL_2}$.

We will now prove this for our setting, considering two random partitions $P_1$ and $P_2$.

Xue et al. (2023) shows the $L_{SSL}$ can be rewritten as $L_{SSL} = -Tr(2M\theta\theta^T) + Tr(M'\theta^T\theta M'\theta^T\theta)$

where $M = \frac{1}{m|B|} \sum_{i=1}^{m|B|} x_i x_i^T$, $M' = \frac{1}{|B|} \sum_{i=1}^{|B|} \left(\frac{1}{m} \sum_{x \in \mathcal{A}(x_i)} x\right)\left(\frac{1}{m} \sum_{x \in \mathcal{A}(x_i)} x\right)^T$ where $m$ is the number of augmentations sampled.

From Xue et al. (2023), we also have $\nabla_\theta(L_{SSL}) = -4\theta M + 4\theta M'\theta^T\theta M'$

With $m$ i.e. number of augmentations large enough, $M = M'$.

We now refer to $M$ for $P_1$ as $M_1$ and $M$ for $P_2$ as $M_2$.

From the construction of $P_1$, we have $M_1 = I$ and from the construction of $P_2$, we have $[M_2]_{i,j} = \delta_{i,k}\delta_{j,k}$ where $\delta_{i,j} = \mathbf{1}[i = j]$ for batch where examples from class $k$ are selected.

From this, we can see that, the $\nabla_\theta(L_{SSL})$ is not equal for $P_1$ and $P_2$.

Hence, we can conclude that $\theta_{SSL_1} \neq \theta_{SSL_2}$ $\qquad\square$

## E    EXAMPLES FROM MKDTAND KRR-ST (LEE ET AL., 2023)

Here, we show examples from CIFAR10, CIFAR100 and TinyImageNet for the data distilled using MKDTand KRR-ST (Lee et al., 2023).

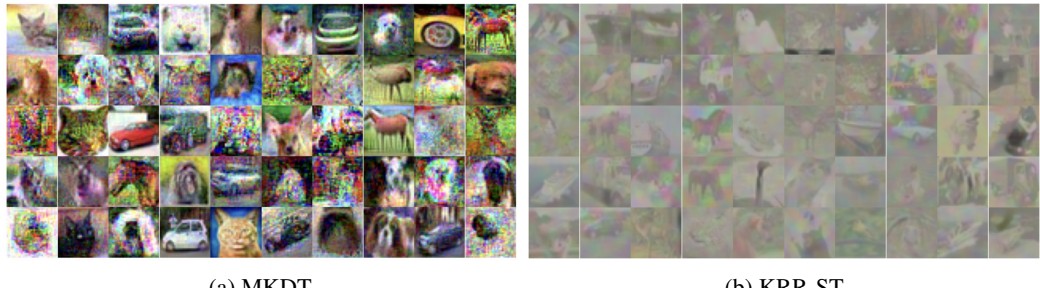

(a) MKDT                                                  (b) KRR-ST

Figure 2: Examples of Distilled Images for CIFAR10

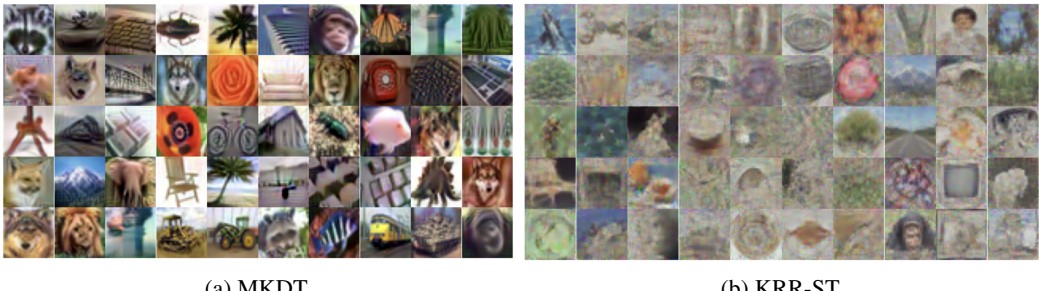

(a) MKDT                                                  (b) KRR-ST

Figure 3: Examples of Distilled Images for CIFAR100

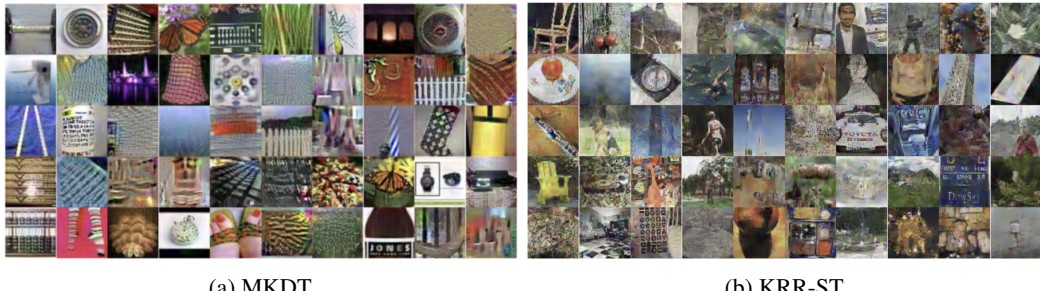

(a) MKDT                                                  (b) KRR-ST

Figure 4: Examples of Distilled Images for TinyImageNet

## F    DM AND MTT-SSL EXPERIMENTS

We also compared our methods with two DD methods adapted to SSL. Specifically, we focused on distilling 5% of CIFAR 100. The results all show that those methods do not outperform MKDT.

To adapt DM for SSL, we used the same codebase and hyperparameters as DM (Zhao & Bilen, 2023). However, during distillation, instead of DM's original approach of sampling random images from a specific class, we modified the process to sample random images from the entire dataset. This adjustment allows us to approximate the behavior of running DM in a label-free setting. The results are shown in 11.

To adapt MTT to SSL, we modified the supervised learning algorithm in getting the trajectory and training the network in the inner loop of the distillation to be the SimCLR SSL algorithm. Due to the

high variance of SSL as analyzed in 4.1, distillation fails to converge, and the poor results further shows that MTT with SSL does not work. The results are shown in 12.

| Dataset | CIFAR100 | CIFAR10 | TinyImageNet | Aircraft | CUB2011 | Dogs | Flowers |
|---|---|---|---|---|---|---|---|
| CIFAR10 (1% Labels) | $7.09_{\pm 0.16}$ | $35.98_{\pm 0.98}$ | $2.43_{\pm 0.19}$ | $2.15_{\pm 0.21}$ | $1.19_{\pm 0.13}$ | $1.71_{\pm 0.18}$ | $1.60_{\pm 0.17}$ |
| CIFAR10 (5% Labels) | $14.77_{\pm 0.10}$ | $46.41_{\pm 0.35}$ | $5.11_{\pm 0.32}$ | $4.94_{\pm 0.88}$ | $1.57_{\pm 0.31}$ | $2.53_{\pm 0.13}$ | $3.10_{\pm 0.33}$ |
| CIFAR100 (1% Labels) | $8.74_{\pm 0.35}$ | $37.75_{\pm 0.87}$ | $2.75_{\pm 0.22}$ | $2.51_{\pm 0.08}$ | $1.20_{\pm 0.11}$ | $2.07_{\pm 0.20}$ | $2.53_{\pm 0.27}$ |
| CIFAR100 (5% Labels) | $17.01_{\pm 0.37}$ | $47.32_{\pm 0.38}$ | $6.40_{\pm 0.21}$ | $5.08_{\pm 0.82}$ | $1.99_{\pm 0.13}$ | $2.77_{\pm 0.28}$ | $4.62_{\pm 0.28}$ |

Table 11: Results for Distilling with DM with SSL.

| Dataset | CIFAR100 | CIFAR10 | TinyImageNet | Aircraft | CUB2011 | Dogs | Flowers |
|---|---|---|---|---|---|---|---|
| CIFAR10 (1% Labels) | $4.63_{\pm 0.30}$ | $22.06_{\pm 1.86}$ | $1.11_{\pm 0.16}$ | $1.92_{\pm 0.26}$ | $0.77_{\pm 0.11}$ | $1.38_{\pm 0.28}$ | $1.26_{\pm 0.37}$ |
| CIFAR10 (5% Labels) | $7.10_{\pm 0.43}$ | $29.04_{\pm 0.58}$ | $1.90_{\pm 0.17}$ | $2.81_{\pm 0.60}$ | $0.88_{\pm 0.08}$ | $1.82_{\pm 0.19}$ | $1.42_{\pm 0.53}$ |
| CIFAR100 (1% Labels) | $4.26_{\pm 0.34}$ | $26.06_{\pm 1.58}$ | $1.20_{\pm 0.14}$ | $1.49_{\pm 0.28}$ | $0.87_{\pm 0.26}$ | $1.24_{\pm 0.07}$ | $1.41_{\pm 0.27}$ |
| CIFAR100 (5% Labels) | $9.66_{\pm 0.24}$ | $38.16_{\pm 0.57}$ | $2.11_{\pm 0.19}$ | $2.83_{\pm 0.41}$ | $1.06_{\pm 0.04}$ | $1.74_{\pm 0.15}$ | $2.38_{\pm 0.19}$ |

Table 12: Results for Distilling with MTT using High Variance SSL Trajectories.

## G    SOCIETAL IMPACT

Our work democratizes the development of models trained with SSL pre-training, by reducing the volume of data needed to train such models, and thus the costs of training these models. However, the risk with dataset distillation is augmenting the biases of the original dataset.

