# OpenReview forum: "Dataset Distillation via Knowledge Distillation: Towards Efficient Self-Supervised Pre-training of Deep Networks"
_ICLR.cc/2025/Conference — ICLR 2025 Poster_

### Official Review · Reviewer_Juoa · 2024-10-25

**Soundness:** 3
**Presentation:** 3
**Contribution:** 2
**Rating:** 6
**Confidence:** 5

**Summary:**

The authors of the paper deal with applying dataset distillation on unlabeled dataset as part of a self-supervised pre-training. A fundamental observation presented in the paper is that optimizing for dataset distillation while using SSL objective is hard to convergence because of a high gradient variance in the batch. This observation was demonstrated empirically and theoretically under some assumptions. To mitigate the high gradient variance, the authors proposed to use Matching Training Trajectories following knowledge distillation, which leads to a much smooth gradients, allowing a stable dataset distillation optimization and accuracy improvements on several downstream tasks.

**Strengths:**

- In my view, the proposed approach effectively transforms the problem of data distillation in an unlabeled setting (SSL objective) into a supervised approach using knowledge distillation loss.

- The concept of enhancing self-supervised approaches by incorporating dataset distillation is interesting.

- The approach presented in the paper is simple and clearly explained.

- The theoretical and experimental motivations regarding high variance gradients appear reasonable and valid.

**Weaknesses:**

- Line 75: “… datasets distilled with ConvNets can transfer to other larger architectures such as ResNets.” Note that ResNets are also a type of Convolutional network (ConvNet).

- What about considering larger (and more modern) architectures? For example, using larger ResNet architectures instead of just 3-4 convolutional layers.

- Do the authors consider connections to data-free knowledge distillation (DFKD)?

- The authors' theoretical derivation is based on very strict assumptions. Statements such as "as confirmed theoretically above" (line 242) may be overstated.

- The paper uses the Barlow Twins SSL objective. Did the authors try other SSL approaches? It would be useful to know if the observations are generalizable and if the accuracy improvements hold across different SSL algorithms.

- Experiments: the authors only compare linear probes on 1% and 5% of the downstream labeled data. Why not present results for additional subset sizes? Especially when comparing to SAS, which is a data pruning method where the main focus is on higher subset sizes (e.g., SAS provides results for 20% subset size and above). This raises the question of whether the comparison to SAS is fair.

**Questions:**

- My main question is more general, concerning the concept of dataset distillation (DD). Despite over four years of research on DD, progress appears to be limited, with most work focused on small datasets and controlled settings. I would appreciate the authors' insights on the future potential and development of dataset distillation.

- I think the authors may consider presenting some connections to the field of data-free knowledge distillation. What do the authors think?

---

> ### Author Response · Authors · 2024-11-22
>
> We thank the reviewer for acknowledging the strengths of our work, namely: 1) the innovative solution to transform dataset distillation (DD) for SSL into a supervised approach using the knowledge distillation loss 2) tackling the important and under-explored problem of DD for SSL 3) simple and clear writing 4) theoretical and empirical motivations for high variance of gradients of SSL.
>
> **W1**: Corrected in revision.
>
> **W2**: In Table 5, we have already included results showing our method can be used to train models as large as ResNet-18.
>
> **W3**: Discussed in Q2
>
> **W4**: Changed in revision to "as illustrated theoretically in the simplified setting above".
>
> **W5**: In Table 6, we have already included results for MKDT applied to SimCLR (another SSL approach).
>
> **W6**: In Table 7, we have already included results evaluating the encoders trained using MKDT distilled sets with larger fractions of labeled data (10%,50%). We mainly focused on very small subsets in MKDT, since dataset distillation has additional benefits over SAS (subset selection for contrastive learning). SAS mainly focuses on reducing training time, while preserving accuracy; whereas dataset distillation is useful to generate extremely small subsets to enable very fast training/adaptation on *memory-limited edge devices*. Moreover, dataset distillation has the additional orthogonal goal of preserving privacy.
>
> **Q1**: In supervised learning, very recent work of [1] has shown promise in generalizing to larger networks and larger datasets (e.g. training a ResNet on a distilled version of ImageNet to 34% accuracy using only 50 images per class). This, however, relies heavily on labels and cannot be applied to dataset distillation for SSL. Currently, dataset distillation for SSL is still underexplored. Our contribution sets the groundwork for how to perform dataset distillation for SSL. As happened with SL dataset distillation, we hope that subsequent work extend this to larger scale models and dataset sizes.
>
> **Q2**: Recent work in distribution matching [40 from paper, 47 from paper, 1] that has achieved remarkable results has relied on data-free knowledge distillation. We did mention these works in our related work for distribution matching methods. The key idea from data-free distillation works of optimizing the synthetic data to match data statistics such as mean, variance etc. [2] has proved extremely helpful in dataset distillation for SL. However, as shown in [40 from paper, 47 from paper, 1], this still requires labels to prevent a collapse of synthetic data from different classes. Nonetheless, the connection to data-free knowledge distillation is indeed interesting and we believe future work could adapt DFKD for DD for SSL, using our idea of converting SSL to SL using knowledge distillation.
> We have also added additional references to DKFD literature to our revision.
>
> We are eager to engage in further discussion to resolve any other comments / concerns.
>
> **References:**
>
> [1] Shao, Shitong, et al. "Generalized large-scale data condensation via various backbone and statistical matching." Proceedings of the IEEE/CVF Conference on Computer Vision and Pattern Recognition. 2024.
>
> [2] Lopes, Raphael Gontijo, Stefano Fenu, and Thad Starner. "Data-free knowledge distillation for deep neural networks." arXiv preprint arXiv:1710.07535 (2017).

---

> > ### Author Response · Authors · 2024-11-25
> >
> > As the discussion period is coming to an end soon, we're hoping to hear if our rebuttal addressed the reviewer's concerns and if we can provide any more clarifications about our work.
> >
> > Thank you once again for your efforts reviewing our paper!

---

> > > ### Author Response · Authors · 2024-11-28
> > >
> > > As the extended discussion period will end in a few days, we're hoping to hear if our rebuttal addressed the reviewer's concerns and if we can provide any more clarifications about our work.
> > >
> > > Thank you once again for your efforts reviewing our paper!

---

> > > > ### Comment · Reviewer_Juoa · 2024-11-30
> > > >
> > > > Thanks for the authors' responses. After reviewing the response and reading the other reviewr's comments, I would like to keep my score.

---

> > > > > ### Author Response · Authors · 2024-12-03
> > > > >
> > > > > Thank you for taking the time to go through our rebuttal!

---

### Official Review · Reviewer_SPPk · 2024-10-29

**Soundness:** 2
**Presentation:** 2
**Contribution:** 3
**Rating:** 8
**Confidence:** 4

**Summary:**

This paper introduces one of the first approaches to dataset distillation for self-supervised learning (SSL) pre-training, addressing the challenge of generating compact, synthetic datasets that can efficiently pre-train deep networks without labeled data. By leveraging a teacher-student framework, the authors demonstrate that using knowledge distillation (KD) significantly reduces the variance in training trajectories, a common issue in SSL due to high variance in gradient updates. This approach, termed "Matching Knowledge Distillation Trajectories" (MKDT), trains a smaller student model to match the embeddings of a larger teacher model, resulting in a low-variance objective that allows effective dataset distillation for SSL. Experimental results indicate that MKDT outperforms prior methods by up to 13% across various downstream tasks with limited labeled data, underscoring its potential for memory- and compute-efficient SSL pre-training​

**Strengths:**

1. This work is among the first to address dataset distillation for self-supervised pre-training, providing an innovative approach to generate compact, synthetic datasets that enable efficient pre-training without labeled data.
2. The theoretical motivation for introducing a teacher-student learning model is compelling. By using knowledge distillation, the method effectively reduces the high variance in gradients commonly seen in self-supervised learning objectives, improving upon naive trajectory matching approaches. While I have not throughly checked the the mathematical proofs, but nonetheless, the proofs intuitively seem convincing.
3. The related work section is particularly well-constructed, offering a thorough exploration of the dataset distillation literature and providing valuable insights into the challenges and developments in both supervised and self-supervised settings, positioning this work within the broader context of current research.

**Weaknesses:**

1. **Limited Discussion of Alternative Distillation Paradigms**: The paper presents knowledge distillation (KD) as a novel proxy for enhancing trajectory matching in SSL dataset distillation. However, the authors omit a comparison with other established dataset distillation methods, such as gradient matching or distribution matching. The Related Work section briefly acknowledges these methods but dismisses them based on label dependency. This lack of comparative analysis leaves a gap: could these paradigms outperform trajectory matching if adapted to SSL? Addressing this gap would strengthen the paper by either justifying the choice of trajectory matching for SSL or empirically showing why KD-based approaches are superior.

2. **High Variance in SSL Gradients and Potential Regularization Techniques**: A core motivation for the KD approach is the high variance in SSL gradients, which makes naive trajectory matching ineffective. Yet, the authors do not explore whether regularization techniques, such as Sharpness-Aware Minimization (SAM), could mitigate this variance issue. SAM, which reduces oscillations in model updates by flattening the loss landscape, could be a promising candidate to stabilize SSL gradients. An empirical analysis incorporating such techniques would clarify whether the KD setup is necessary or if regularization alone could suffice.

3. **Clarification and Takeaways in Figure 1**: Figure 1 illustrates the challenges of applying MTT to SSL, emphasizing issues like high gradient variance and chaotic updates to synthetic images. However, the figure captions lack concise takeaways summarizing the implications of each plot. Clearer captions explaining the significance of variance trends and how they justify the proposed KD approach would improve the paper's readability and impact.

**Minor Weakness (Formatting and Page Limit)**: The submission uses a numerical citation format, which does not adhere to the ICLR 2025 requirements of (Author, Year) format. Given the 10-page length, updating the citation style within the limit should be manageable.

**Reference**:
[1] Minimizing the Accumulated Trajectory Error to Improve Dataset Distillation, CVPR 2023.

**Questions:**

1. In Figure 1, what are the primary insights that should be drawn from each sub-figure? Specifically, could the authors clarify how each metric in Figure 1 (e.g., variance in weights, distillation loss) directly impacts the effectiveness of the KD-based trajectory matching approach?

2. The paper argues for the superiority of trajectory matching in the SSL setting. Could the authors elaborate on any specific challenges or limitations of gradient matching and distribution matching that make them unsuitable for SSL dataset distillation, as compared to trajectory matching? (this question ties to one of the points in the weakness section)

3. Regarding the high-variance problem in SSL gradients, has there been an investigation into alternative approaches, such as sharpness-aware minimization, to reduce gradient variance? What led to the decision to focus solely on KD rather than considering variance-reducing techniques as a complementary approach? (also raised in the weakness section)

4. In Table 2, could the authors clarify the relationship between initialization choice (e.g., high-loss vs. random subsets) and performance? Is there a grounded rationale behind the preference for high-loss subset initialization?

5. Could the authors provide more context on how the choice of SSL algorithm (e.g., SimCLR vs. Barlow Twins) affects the efficacy of the MKDT method? Is there a fundamental reason one algorithm would be preferred over another in the KD-based trajectory matching framework? Which method among SimCLR and BarlowTwins as faster convergence while learning the distilled samples? Can this method also be extended to other SSL methods such as masked reconstruction?

6. Lastly, in the methodology, could the authors clarify how they determined the optimal values for hyperparameters like the number of distillation steps, K expert trajectories, and distillation loss thresholds?

---

> ### Author Response · Authors · 2024-11-22
>
> We thank the reviewer SPPk for their appreciation of 1) the innovative approach we provide for the important problem of dataset distillation for SSL 2) the theoretical motivation for using knowledge distillation to generate lower variance trajectories to enable trajectory matching for DD for SSL 3) the well-written related works section, situating the contribution of our work in the current research.
>
> We now address the weaknesses and questions raised by the reviewer:
>
> **W1**: As the reviewer observed, in our related work we did identify the label dependency of the SL method as preventing them from being applied to SSL. Gradient matching methods necessarily need labels, as it is crucial to match gradients **per-class** to learn class-discriminative features as shown in [46 from paper]. For distribution matching, a similar concern exists where distilling without labels would lead to a collapse in representations i.e. all classes would have nearly identical representations. We confirm this empirically in the table below, showing that DM without labels is not able to distill anything meaningful, performing no better than random subsets. Since MTT is the only method that can indeed distill without labels and still learn class-discriminative features, we do believe MTT is the best candidate from SL distillation techniques to be adapted to SSL.
>
> **Table: Distribution Matching (DM) without Labels (Distilled Data Size = 5%)**
>
> The fraction % next to the dataset name denotes the fraction of downstream labels used.
> (Compare this to #s in Table 4 from paper)
> | Dataset   | CIFAR100    | CIFAR10     | TinyImageNet | Aircraft    | CUB2011     | Dogs        | Flowers     |
> |-----------|-------------|-------------|--------------|-------------|-------------|-------------|-------------|
> | C10 1%    | 7.09 ± 0.16 | 35.98 ± 0.98| 2.43 ± 0.19  | 2.15 ± 0.21 | 1.19 ± 0.13 | 1.71 ± 0.18 | 1.60 ± 0.17 |
> | C10 5%    | 14.77 ± 0.10| 46.41 ± 0.35| 5.11 ± 0.32  | 4.94 ± 0.88 | 1.57 ± 0.31 | 2.53 ± 0.13 | 3.10 ± 0.33 |
> | C100 1%   | 8.74 ± 0.35 | 37.75 ± 0.87| 2.75 ± 0.22  | 2.51 ± 0.08 | 1.20 ± 0.11 | 2.07 ± 0.20 | 2.53 ± 0.27 |
> | C100 5%   | 17.01 ± 0.37| 47.32 ± 0.38| 6.40 ± 0.21  | 5.08 ± 0.82 | 1.99 ± 0.13 | 2.77 ± 0.28 | 4.62 ± 0.28 |
>
>
> **W2, W3**: We combined the responses for these 2 weaknesses since a clear explanation of Figure 1 helps us explain why other variance reduction techniques are unlikely to be sufficiently effective. Firstly, as shown in [1], SAM is a regularization for finding more-generalizable minima by penalizing the sharpness. This is distinct from the problem of reducing variance of gradients. In fact [5] shows that SAM’s optimization leads to higher variance of gradients and can benefit significantly itself from variance reduction. Secondly, classical variance reduction techniques have been shown to be unsuccessful for deep neural networks [4]. Thirdly, we provide additional results (see Table ) showing that MTT applied directly to higher variance SSL trajectories leads to distilled sets that perform worse than even trivial baselines (random subsets / no-pretraining).
> Additionally, in Figure 1, we do indeed explore a simple alternative to reduce variance for SSL, i.e., increasing the batch size, and show how this is still insufficient:
> In Figure 1a, we provide evidence for the higher variance in gradients using empirical estimates. Since weights at the end of each iteration are determined exactly by the gradient, we use the variance of the weights at end of each iteration for models starting from the same initialization to estimate this. As we see, the SL (KD) loss has far lower variance than SSL. Moreover, while increasing the batch size by 4x does reduce the variance slightly, it is not nearly as much as SL. This indicates that reducing the variance of SSL through other regularization techniques may also likewise be insufficient.
> Figure 1b shows that as a result of the higher variance, when the distillation process optimizes the distilled set to match the training trajectories, it is not able to successfully optimize the distilled set to match the trajectories i.e. it cannot minimize the distillation loss. This is due to how challenging this optimization is due to the high variance of SSL.
> Finally, Figure 1c confirms that the insufficient minimization of distillation loss is problematic. In particular, it shows that for the cases where distillation loss was insufficiently minimized, there is little to no change between distilled images and their initializations indicating that the distillation has been unsuccessful.
>
> **Q1**: Answered with W2/W3.
>
> **Q2**: Answered with W1.
>
> **Q3**: Answered with W2/W3.

---

> ### Author Response · Authors · 2024-11-22
>
> **Q4**: We empirically showed that, on most tasks, initializing with the high-loss subset performs slightly better than initializing with a random subset. Examples with a high loss correspond to more ambiguous data points likely to lie on the boundary of the latent classes of the pretraining data (high loss images from CIFAR100, shown in https://anonymous.4open.science/r/iclr_mkdt_rebuttal-2DE1/iclr_rebuttal_fig.png). As a result, initializing the distilled set with these high-loss examples allows the distilled set to better learn representations of boundary examples, leading the encoder to more closely preserve the teacher model’s representations on them (see average MSE on the top 1% of high-loss examples in https://anonymous.4open.science/r/iclr_mkdt_rebuttal-2DE1/iclr_rebuttal_fig.png). Consequently, since the boundary points are represented more accurately, the linear classifier trained on these representations with downstream data achieves higher accuracy. This is not central to our method but rather an additional component that provides further performance improvements at no extra cost.
>
> **Q5**: MKDT does indeed enable dataset distillation for **various SSL methods** (including masked reconstruction) since the only requirement is **representations from a teacher model trained with SSL**. As we show in Table 6, we have results extending MKDT to SimCLR as well, showing significant gain over baselines. We believe that the effectiveness of the distilled set will be determined by how effective the given SSL algorithm is training the teacher model.
>
> **Q6**: Largely we borrowed hyperparameters from MTT [1 from paper]. In particular, we use nearly identical number of synthetic steps (40 for us v/s 30-50 for [1 from paper] on CIFAR10/100 and 10 for both on TinyImageNet) and expert epochs (2 epochs, across all datasets, for both us and [1 from paper]). As seen in [1 from paper], when distillation is more challenging, e.g., on larger datasets such as TinyImageNet, the max start epoch is reduced in order to only distill early training dynamics and make optimization more tractable. Even with knowledge distillation trajectories, we observed that optimizing the distillation loss for SSL is harder (since we deal with the more difficult problem of supervised regression with unique representations as labels), thus we reduced the max start epoch by ~10x (2 for us  v/s 20 for [1 from paper] on CIFAR10/100 and 2 for us v/s 10 [1 from paper] on TinyImageNet).
>
> We are eager to engage in further discussion to resolve any other concerns or comments!
>
> **References:**
>
> [1] Foret, Pierre, et al. "Sharpness-aware minimization for efficiently improving generalization." arXiv preprint arXiv:2010.01412 (2020).
>
> [2] Chen, Xuxi, et al. "Data distillation can be like vodka: Distilling more times for better quality." Proceedings of the International Conference on Learning Representations (ICLR), 2024.
>
> [3] Yang, Yu, Hao Kang, and Baharan Mirzasoleiman. "Towards sustainable learning: Coresets for data-efficient deep learning." International Conference on Machine Learning. PMLR, 2023.
>
> [4] Defazio, Aaron, and Léon Bottou. "On the ineffectiveness of variance reduced optimization for deep learning." Advances in Neural Information Processing Systems 32 (2019).
>
> [5] Li, Bingcong, and Georgios Giannakis. "Enhancing sharpness-aware optimization through variance suppression." Advances in Neural Information Processing Systems 36 (2024).

---

> > ### Comment · Reviewer_SPPk · 2024-11-23
> > **Response to the Authors**
> >
> > Thank you for your detailed and thoughtful response to the initial review. I appreciate the effort you have put into addressing the concerns and providing clear explanations and justifications for the choices made in your work. Given that all my concerns have been addressed satisfactorily, I am happy to increase my score for this submission.

---

> > > ### Author Response · Authors · 2024-11-25
> > >
> > > We're glad you found our rebuttal **detailed** and **thoughtful**, and that the **clear explanations** we provided for choices made in our work helped address your concerns.
> > >
> > > Thank you once again for taking the time to review our paper so thoughtfully!

---

### Official Review · Reviewer_1SyT · 2024-11-03

**Soundness:** 3
**Presentation:** 3
**Contribution:** 2
**Rating:** 8
**Confidence:** 4

**Summary:**

- This paper propose a data distiallation method for self-supervised pretraining. This methods MKDT is a based on knowledge distiallation method, which is mainly motivated by the study on the gradient's variance of different loss function.

**Strengths:**

- The first, as this paper claims, effective data distillation method for SSL pretraining is proposed.
- Outstanding experimental performance on provided datasets, \textit{e.g.}, CIFAR-10, 100/Tiny ImageNet.

**Weaknesses:**

- Not matched theory and implementation. The toy case in theoretical analyses is too simple and the gap between analyses and implementation is too large.
    1. The linear model is provided in the main paper only. However, even in linear probe experiments, the tested model is non-linear. Moreover, in the literature of the related ones (\textit{e.g.}, sharp-aware generalization, bias-variance trade-off, Out-of-Distribution Generalization), analyses about gradient's variance on different loss design are usually conducted on non-linear case (\textit{e.g.}, 2-layer network with non-linear Activation function, such as ReLU in early years).[1,2,3]
    2. The derivative and presentation of its proof is too complicated however the logic is actually trival (i.e., simply unfold the deviation of the gradient of SL and SSL, and then compare each term by \textit{easy to see is larger}).
- Pretraining is important nowadays. However, the model architecture is too small and simple, for which pretraining is not as important as the large models (\textit{e.g.}, LLMs), which is not discussed in the main paper.
    1. In the assumption of the theorem, citation [4] about sparse code is on multi-modal pretraining, but similar architecture is not mentioned in the rest of the paper.
- Experiments about generalization is confusing.
    1. In the distillation setup, it claims that the teacher model is ResNet-18 and the student model is ConvNet. However, the models in the experiments about generalization to Larger architecture are ResNet-18 and ResNet-10. The models are not larger at all, and ResNet-L (L>18) are available in the provided code.
- Typos and inconsistent prensentation, \textit{e.g.}, synhronous in Line 215, pre-training and pretraining.

[1] Estimating Example Difficulty using Variance of Gradients. CVPR 2022
[2] Fishr: Invariant Gradient Variances for Out-of-Distribution Generalization. ICML 2022
[3] Rethinking Bias-Variance Trade-off for Generalization of Neural Networks. ICML 2022
[4] Data-efficient contrastive language-image pretraining: Prioritizing data quality over quantity. AISTATS 2024

**Questions:**

It seems like that the logic of the paper is: the gradient variance of SSL is larger than the one of SL, thus the knowledge distillation is needed to lower the variance due to the relationship between gradient and trajectory matching. The questions are:
- It's due to the high variance of the SSL gradient, why not show and print the empirical gradient variance directly?
- How is the straightforward relationship between model performance and gradient variance, if albation study on trajectory sampling is considered?

I first give 6 here, actually, I tend to give 5.5. therefore, some important concerns should be discussed during the rebuttal period.

---

> ### Author Response · Authors · 2024-11-22
>
> We’d like to thank reviewer 1SyT for appreciating our work’s contribution as 1) the first *effective* dataset distillation method for SSL pre-training, as well as 2) our approach’s outstanding experimental performance on CIFAR-10, CIFAR100 and TinyImageNet.
>
> We now address the weaknesses and questions raised by the reviewer:
>
> **W1**: We included a simplified theoretical analysis to highlight how the interaction between examples in a batch by SSL loss leads to higher gradient variance. Thus, we unrolled all the terms to illustrate how the variance for SSL gradient would be larger. The works cited by the reviewer are all for supervised learning; we are not aware of existing analysis of the variance of gradients with non-linear models for contrastive learning. Due to the more complicated nature of the SSL/CL loss, prior works in CL theory [1,2,3] have also relied on analyses of linear models and have demonstrated how such analysis corresponds with the behavior of deep nonlinear networks. Providing an expression for variance of SSL gradient in a more general setting is a challenging problem in its own right. This is because **gradient of each example depends on every other example in the batch**, hence introducing additional complications. Such an analysis is beyond the scope of this work. Nonetheless, we have revised our claims from “confirmed theoretically” to “illustrate theoretically, in a simplified setting” to reflect the gap between theory and implementation.
>
> **W2**: Although we use 4-5 layer ConvNets for distillation, in Table 5, we do demonstrate that datasets distilled with these smaller ConvNets can be used to pre-train models as large as ResNet-18. This pre-training is effective as it shows generalization to a variety of downstream tasks with only 1-5% of data, when evaluated by training only a linear classifier on the representations of the pretrained ResNets. We note that using 4-5 layer ConvNets is also common practice in dataset distillation for SL [1, 23, 25, 42, 46, 48 etc. from paper]. This is because gradient, weight, or distribution matching with networks larger than ConvNets becomes prohibitive due to the very high cost and difficulty of optimization. Enabling further scaling of this method to larger networks is indeed an interesting direction for future research. Recent work [6, 9 from paper] on improving the scalability and optimization of trajectory matching, orthogonal to our contribution, can be useful to this end. With regards to the citation for sparse-coding model, we have replaced them with citations from prior literature on unimodal contrastive learning.
>
> **W3**: There might be a misunderstanding of the goal of our experiment here. We are not trying to demonstrate generalization to architectures larger than the “teacher network”. Instead, our goal is to demonstrate that our methodology can apply to networks larger than the 4-5 layer ConvNets used for distillation. The results on ResNet-10 and ResNet-18 indeed prove that this methodology can generalize to larger networks.
>
> **W4**: Thank you for catching these! We've addressed them in our revision.
>
> **Q1**: Since weights = lr * gradient, our Figure 1 shows the high variance of gradients using the variance of weights, after each iteration. The first step shows exactly the high variance of gradients and subsequent steps shows how this leads to even further increase in variance of weights over iterations. Moreover, we focus on showing the impact of the high variance of gradients on variance of weights since trajectory matching matches weights, at different iterations, rather than gradients. In Figure 1a, we compute the variance, in weights, after each iteration (step), across 5-models starting from the same initialization, for both SL (in particular, the knowledge distillation loss which is a supervised regression loss) and SSL. The figure confirms the smaller variance of our method (denoted as MTT SL) vs MTT SSL.

---

> ### Author Response · Authors · 2024-11-22
>
> **Q2**: First, we provide additional results (see https://openreview.net/forum?id=c61unr33XA&noteId=am03RRoWCi) showing that MTT applied directly to higher variance SSL trajectories leads to distilled sets that perform worse than even trivial baselines (random subsets / no-pretraining). Second, as discussed earlier, Fig 1 confirms the connection between variance and distillation performance. Figure 1a established that the variance of the gradient is large for SSL as compared to SL. Figure 1b shows that as a result, distillation loss cannot be effectively minimized by MTT on higher variance trajectories. Figure 1c confirms that the inability to minimize distillation loss leads to no change in the images from their initialization - indicating no distillation has occurred. To further confirm this connection, we considered reducing variance for SSL by increasing the batch size (denoted by 4x batch size SSL). We confirm in Figure 1a that this does indeed reduce the variance. Figure 1b shows that this leads to a slightly better minimization in distillation loss and Figure 1c shows that this leads to a slightly more effective distillation with images changing from their initialization. Overall, the distilled images using the high variance SSL trajectories are nearly identical to the real images used for initialization. Thus, they offer no benefits to pre-training, privacy etc. We use these results together to confirm empirically the connection between gradient variance and effectiveness of trajectory matching distillation.
>
> We are eager to engage in further discussion to resolve any other concerns or comments!
>
> **References**:
>
> [1] Ji, Wenlong, et al. "The power of contrast for feature learning: A theoretical analysis." Journal of Machine Learning Research 24.330 (2023): 1-78.
>
> [2] Xue, Yihao, et al. "Investigating the Benefits of Projection Head for Representation Learning." Proceedings of the International Conference on Learning Representations (ICLR), 2024.
>
> [3] Xue, Yihao, et al. "Which features are learnt by contrastive learning? On the role of simplicity bias in class collapse and feature suppression." International Conference on Machine Learning. PMLR, 2023.

---

> > ### Author Response · Authors · 2024-11-25
> >
> > As the discussion period is coming to an end soon, we're hoping to hear if our rebuttal addressed the reviewer's concerns and if we can provide any more clarifications about our work.
> >
> > Thank you once again for your efforts reviewing our paper!

---

> > > ### Author Response · Authors · 2024-11-28
> > >
> > > As the extended discussion period will end in a few days, we're hoping to hear if our rebuttal addressed the reviewer's concerns and if we can provide any more clarifications about our work.
> > >
> > > Thank you once again for your efforts reviewing our paper!

---

> > > > ### Comment · Reviewer_1SyT · 2024-12-02
> > > >
> > > > Thx for the rebuttal. It solves most concerns. I raised the score. Good luck!
> > > > citing more recent dd papers is good for this work.

---

> > > > > ### Author Response · Authors · 2024-12-03
> > > > >
> > > > > Thank you for taking the time to go through our rebuttal!

---

### Official Review · Reviewer_YkVh · 2024-11-04

**Soundness:** 3
**Presentation:** 3
**Contribution:** 3
**Rating:** 6
**Confidence:** 4

**Summary:**

This paper explores dataset distillation methods for self-supervised learning (SSL). The authors demonstrate that the MTT method cannot be directly applied to SSL due to high trajectory variance, both theoretically and empirically. To address this, they propose a solution leveraging knowledge distillation (KD) to reduce the length and variance of SSL trajectories.

**Strengths:**

- Self-supervised dataset distillation is an important yet underdeveloped area with wide-ranging applications.
- The empirical and theoretical evidence provided (Theorem 4.1 and Figure 1) for high gradient and trajectory variance of MTT in SSL settings is convincing and insightful.

**Weaknesses:**

- The performance improvements over random subset and high-loss subset are marginal in many cases, especially on the larger TinyImageNet dataset (Table 3). This raises concerns about the method’s practical value, as it requires significant computational resources to distill synthetic data while yielding minimal improvement.
- The absence of KRR-ST results in Tables 4, 5, 6, and 7 limits the ability to assess the proposed method's effectiveness, given that KRR-ST is a closely related baseline.
- The MKDT method appears to add only a knowledge distillation process to MTT, where a student model mimics the SSL teacher model’s representations. The synthetic data is then learned by matching the student model’s trajectory, rather than the teacher model’s trajectory, as in MTT. It is unclear why introducing a student model as an intermediary reduces trajectory variance, or what specific role the student network plays throughout the data distillation process. The authors are encouraged to provide further discussion or insights into this mechanism.

**Questions:**

- Line 295: Incorrect format in “[41] trains student ….”
- The MKDT method uses ResNet-18 as the teacher model and ConvNet as the student model. What is the backbone network used for KRR-ST in Tables 2 and 3?
- Presenting some distilled images would help demonstrate the proposed method’s effectiveness and its advantage over the baseline KRR-ST.

---

> ### Author Response · Authors · 2024-11-22
>
> We’d like to thank the reviewer YkVh for appreciating 1) the importance of the problem we tackle i.e. dataset distillation for self-supervised learning and 2) the empirical and theoretical evidence we provide for high variance of SSL trajectories that prevent successful application of trajectory matching (MTT).
>
> We now address the weaknesses and questions raised by the reviewer:
>
> **W1**: Our evaluation measures performance across several downstream datasets and We have significant improvement when pre-training on CIFAR10/CIFAR100, up to 13%. We acknowledge that, for TinyImageNet, on some datasets, the improvement over subsets is smaller. However, this is not a limitation of our method which proposes using knowledge distillation (KD) trajectories to enable data distillation for SSL. Instead, this is a limitation inherited from trajectory matching (MTT) for higher-resolution datasets such as TinyImageNet. As seen in the MTT paper [1 from paper], the improvement over random subsets is far smaller for higher resolution datasets such as TinyImageNet as compared to CIFAR10/CIFAR100. Remedying this problem for both trajectory matching for SL and SSL is an interesting direction for future work.
>
> **W2**: We have added these results in our revision (to Tables, 4,5,6 and 7). We also include these results below. As we see, across all these settings, our proposal, MKDT, continues to outperform KRR-ST significantly.
>
> For all the tables below, % in brackets indicates the % of labeled downstream data available.
>
> **Table 4: KRR-ST 5%**
> | Dataset   | CIFAR100    | CIFAR10     | TinyImageNet | Aircraft    | CUB2011     | Dogs        | Flowers     |
> |-----------|-------------|-------------|--------------|-------------|-------------|-------------|-------------|
> | CIFAR10 1%    | 8.69 ± 0.32 | 36.69 ± 0.88| 3.20 ± 0.23  | 2.26 ± 0.13 | 1.33 ± 0.09 | 1.91 ± 0.34 | 2.39 ± 0.18 |
> | CIFAR10 5%    | 16.95 ± 0.53| 47.40 ± 0.34| 7.10 ± 0.27  | 5.56 ± 0.77 | 1.98 ± 0.07 | 2.78 ± 0.16 | 4.38 ± 0.04 |
> | CIFAR100 1%   | 9.02 ± 0.24 | 37.86 ± 1.14| 2.94 ± 0.13  | 2.42 ± 0.35 | 1.50 ± 0.07 | 1.99 ± 0.19 | 3.04 ± 0.36 |
> | CIFAR100 5%   | 17.24 ± 0.47| 47.53 ± 0.11| 6.60 ± 0.32  | 5.37 ± 0.85 | 2.31 ± 0.33 | 2.87 ± 0.27 | 5.23 ± 0.14 |
> | TinyImageNet 1%   | 7.54 ± 0.35 | 34.27 ± 1.36| 3.19 ± 0.22  | 2.11 ± 0.23 | 1.30 ± 0.12 | 1.68 ± 0.20 | 2.65 ± 0.64 |
> | TinyImageNet 5%   | 13.71 ± 0.30| 42.82 ± 0.46| 6.50 ± 0.23  | 4.36 ± 0.49 | 1.97 ± 0.06 | 2.75 ± 0.37 | 3.97 ± 0.14 |
>
> **Table 5: ResNet10**
> | Dataset   | CIFAR100    | CIFAR10     | TinyImageNet | Aircraft    | CUB2011     | Dogs        | Flowers     |
> |-----------|-------------|-------------|--------------|-------------|-------------|-------------|-------------|
> | KRR-ST 5%  | 13.84 ± 0.78| 39.21 ± 0.55| 8.04 ± 0.52  | 2.12 ± 0.15 | 1.16 ± 0.05 | 1.77 ± 0.14 | 4.56 ± 0.42 |
>
> **Table 5: ResNet18**
> | Dataset   | CIFAR100    | CIFAR10     | TinyImageNet | Aircraft    | CUB2011     | Dogs        | Flowers     |
> |-----------|-------------|-------------|--------------|-------------|-------------|-------------|-------------|
> | KRR-ST 5%  | 12.30 ± 0.83| 35.73 ± 1.07| 7.21 ± 0.35  | 2.32 ± 0.39 | 1.18 ± 0.16 | 1.81 ± 0.14 | 2.45 ± 0.12 |
>
> **Table 6: KRR-ST SimCLR**
> | Dataset   | CIFAR100    | CIFAR10     | TinyImageNet | Aircraft    | CUB2011     | Dogs        | Flowers     |
> |-----------|-------------|-------------|--------------|-------------|-------------|-------------|-------------|
> | C10 1%    | 8.38 ± 0.17 | 36.90 ± 1.30| 2.95 ± 0.12  | 2.45 ± 0.13 | 1.19 ± 0.09 | 1.87 ± 0.18 | 2.35 ± 0.06 |
> | C10 5%    | 16.29 ± 0.37| 46.87 ± 0.52| 6.31 ± 0.43  | 5.31 ± 0.63 | 1.89 ± 0.14 | 2.66 ± 0.18 | 4.36 ± 0.16 |
> | C100 1%   | 8.38 ± 0.36 | 36.57 ± 1.02| 3.01 ± 0.22  | 2.41 ± 0.15 | 1.28 ± 0.02 | 1.71 ± 0.30 | 1.98 ± 0.24 |
> | C100 5%   | 15.75 ± 0.46| 46.76 ± 0.50| 6.17 ± 0.20  | 5.43 ± 0.65 | 1.93 ± 0.06 | 2.61 ± 0.25 | 3.55 ± 0.29 |
>
> **Table 7: KRR-ST 2% Distilled Data with Larger Label Fractions**
> | Dataset   | CIFAR100    | CIFAR10     | TinyImageNet | Aircraft    | CUB2011     | Dogs        | Flowers     |
> |-----------|-------------|-------------|--------------|-------------|-------------|-------------|-------------|
> | C10 10%   | 21.01 ± 0.14| 51.02 ± 0.53| 8.95 ± 0.26  | 7.91 ± 0.76 | 2.44 ± 0.12 | 3.54 ± 0.29 | 6.82 ± 0.64 |
> | C10 50%   | 29.01 ± 0.30| 58.09 ± 0.07| 15.94 ± 0.31 | 17.60 ± 1.04| 5.01 ± 0.34 | 6.81 ± 0.35 | 15.92 ± 0.64|
> | C100 10%  | 21.39 ± 0.16| 52.40 ± 0.73| 8.21 ± 0.07  | 7.64 ± 0.36 | 2.34 ± 0.12 | 3.76 ± 0.25 | 6.52 ± 1.28 |
> | C100 50%  | 29.46 ± 0.81| 58.57 ± 0.78| 15.70 ± 0.07 | 15.89 ± 0.54| 4.82 ± 0.41 | 7.00 ± 0.18 | 15.07 ± 0.46|
> | Tiny 10%  | 17.02 ± 0.26| 45.48 ± 0.84| 8.88 ± 0.41  | 5.29 ± 0.16 | 2.08 ± 0.21 | 3.21 ± 0.15 | 6.10 ± 1.44 |
> | Tiny 50%  | 20.01 ± 0.40| 48.16 ± 1.18| 13.59 ± 0.47  | 8.66 ± 1.29 | 3.46 ± 0.22 | 4.98 ± 0.31 | 15.12 ± 0.37|

---

> ### Author Response · Authors · 2024-11-22
>
> **W3**: We provide additional results (https://openreview.net/forum?id=c61unr33XA&noteId=am03RRoWCi) where we directly apply trajectory matching (MTT) to the  higher variance SSL trajectories and show that the corresponding distilled sets perform worse than even trivial baselines (random subsets / no-pretraining). The key insight of our method is that using the student model as intermediary, we are able to convert the problem from SSL distillation, where the gradient has high variance, to distillation of a **supervised regression** problem where the labels are representations learned by the teacher model. As discussed in Sections 4.1 and 4.2, the supervised regression problem has lower gradient variance as the gradient of each example is independent of other examples in the batch, whereas for SSL, the gradient of each example depends on all other examples in the mini-batch (hence is very sensitive to the choice of mini-batches). The evidence we provide in Thm 4.1 and Figure 1 supports our claims: 1) the variance of the SL regression (knowledge distillation) is less than that of SSL and 2) the lower variance makes distillation more effective.
>
> **Q1**: We have fixed this in our revision.
>
> **Q2**: ResNet-18 trained with BarlowTwins (same as MMKDT) (we follow the exact methodology & hyperparameters specified by KRR-ST and have added this to our Appendix)
>
> **Q3**: We have added examples of distilled images for CIFAR10, CIFAR100 and TinyImageNet for both MKDT and KRR-ST in **Appendix E**. From these, we can clearly observe that MKDT distilled images seem to sharpen the salient class-features of the image, while blurring out background and other extraneous details; in contrast, the images distilled by KRR-ST appear noisy and blurred.
>
> We are eager to engage in further discussion to resolve any other concerns or comments!

---

> > ### Author Response · Authors · 2024-11-25
> >
> > As the discussion period is coming to an end soon, we're hoping to hear if our rebuttal addressed the reviewer's concerns and if we can provide any more clarifications about our work.
> >
> > Thank you once again for your efforts reviewing our paper!

---

> > > ### Comment · Reviewer_YkVh · 2024-11-28
> > > **Response to the rebuttal**
> > >
> > > Thanks for all the efforts on addressing my concerns, I have increased my score.

---

> > > > ### Author Response · Authors · 2024-11-28
> > > >
> > > > Thank you for taking the time to go through our rebuttal!

---

### Author Response · Authors · 2024-11-22
**Results applying MTT directly to High Variance SSL Trajectories (Distilled Dataset Size = 5% of Original Dataset)**

**Table MTT for High Variance SSL Trajectories**
| Dataset   | CIFAR100    | CIFAR10     | TinyImageNet | Aircraft    | CUB2011     | Dogs        | Flowers     |
|-----------|-------------|-------------|--------------|-------------|-------------|-------------|-------------|
| C10 (1% Downstream Labels)    | 4.63 ± 0.30 | 22.06 ± 1.86| 1.11 ± 0.16  | 1.92 ± 0.26 | 0.77 ± 0.11 | 1.38 ± 0.28 | 1.26 ± 0.37 |
| C10 (5%  Downstream Labels)  | 7.10 ± 0.43 | 29.04 ± 0.58| 1.90 ± 0.17  | 2.81 ± 0.60 | 0.88 ± 0.08 | 1.82 ± 0.19 | 1.42 ± 0.53 |
| C100 (1% Downstream Labels)   | 4.26 ± 0.34 | 26.06 ± 1.58| 1.20 ± 0.14  | 1.49 ± 0.28 | 0.87 ± 0.26 | 1.24 ± 0.07 | 1.41 ± 0.27 |
| C100 (5%  Downstream Labels)   | 9.66 ± 0.24 | 38.16 ± 0.57| 2.11 ± 0.19  | 2.83 ± 0.41 | 1.06 ± 0.04 | 1.74 ± 0.15 | 2.38 ± 0.19 |

As confirmed explicitly in this table, applying MTT directly to the high variance SSL trajectories indeed leads to an ineffective distilled set. In fact, training on this distilled set is even worse than Random Subsets (compare with Table 4 in main paper).

---

### Author Response · Authors · 2024-11-22
**Revisions to Submission Highlighted in Blue**

We thank the reviewers for their valuable suggestions to our submission. We've made changes to our submission based on these and highlighted these changes in blue.

---

### Meta-Review · Area_Chair_Y21L · 2024-12-20

**Metareview:**

This paper introduces a novel approach for dataset distillation in self-supervised learning (SSL) pre-training, addressing the inherent challenge of high gradient variance in SSL objectives. The proposed method, Matching Knowledge Distillation Trajectories (MKDT), leverages a teacher-student framework to stabilize gradients and improve the optimization of synthetic datasets. Its innovative approach, strong empirical performance, and theoretical grounding make it a solid candidate for acceptance. The thorough rebuttal further solidified the case for the paper, addressing all reviewer concerns comprehensively.

**Additional Comments On Reviewer Discussion:**

All reviewers agreed to accept this paper.

---

### Decision · Program_Chairs · 2025-01-22

Accept (Poster)